# The 22q11.2 region regulates presynaptic gene-products linked to schizophrenia

Ralda Nehme [1,2,19,20✉], Olli Pietiläinen [1,2,19,20✉], Mykyta Artomov [1,3], Matthew Tegtmeyer[1,2], Vera Valakh[1,2], Leevi Lehtonen[4], Christina Bell[5], Tarjinder Singh [1], Aditi Trehan[1,2], John Sherwood[1,2], Danielle Manning[1], Emily Peirent [1,2], Rhea Malik[2], Ellen J. Guss [2], Derek Hawes[1,2], Amanda Beccard[1], Anne M. Bara[1,2], Dane Z. Hazelbaker[1], Emanuela Zuccaro[2], Giulio Genovese [1], Alexander A. Loboda[1,3,6,7], Anna Neumann [1], Christina Lilliehook[1], Outi Kuismin[8,9,10,11], Eija Hamalainen[4], Mitja Kurki[1,4,8], Christina M. Hultman[12], Anna K. Kähler[12], Joao A. Paulo [5], Andrea Ganna [1], Jon Madison[1], Bruce Cohen [13], Donna McPhie[13], Rolf Adolfsson[14], Roy Perlis[15], Ricardo Dolmetsch[16], Samouil Farhi [1], Steven McCarroll[1], Steven Hyman[1,2], Ben Neale [1], Lindy E. Barrett [1,2], Wade Harper [5], Aarno Palotie [1,4,8,17], Mark Daly [1,3,4,8,17] & Kevin Eggan [1,2,18,20✉]

It is unclear how the 22q11.2 deletion predisposes to psychiatric disease. To study this, we generated induced pluripotent stem cells from deletion carriers and controls and utilized CRISPR/Cas9 to introduce the heterozygous deletion into a control cell line. Here, we show that upon differentiation into neural progenitor cells, the deletion acted in trans to alter the abundance of transcripts associated with risk for neurodevelopmental disorders including autism. In excitatory neurons, altered transcripts encoded presynaptic factors and were associated with genetic risk for schizophrenia, including common and rare variants. To understand how the deletion contributed to these changes, we defined the minimal protein-protein interaction network that best explains gene expression alterations. We found that many genes in 22q11.2 interact in presynaptic, proteasome, and JUN/FOS transcriptional pathways. Our findings suggest that the 22q11.2 deletion impacts genes that may converge with psychiatric risk loci to influence disease manifestation in each deletion carrier.

[1] Stanley Center for Psychiatric Research, Broad Institute of Harvard and MIT, Cambridge, MA 02142, USA. [2] Department of Stem Cell and Regenerative Biology, and the Harvard Institute for Stem Cell Biology, Harvard University, Cambridge, MA 02138, USA. [3] Analytic and Translational Genetics Unit, Department of Medicine, Massachusetts General Hospital, Boston, MA 02114, USA. [4] Institute for Molecular Medicine Finland, University of Helsinki, FI-00014 Helsinki, Finland. [5] Department of Cell Biology, Blavatnik Institute of Harvard Medical School, Boston, MA, USA. [6] ITMO University, St. Petersburg, Russia. [7] Almazov National Medical Research Centre, Saint-Petersburg, Russia. [8] Psychiatric & Neurodevelopmental Genetics Unit, Massachusetts General Hospital, Boston, MA 02114, USA. [9] PEDEGO Research Unit, University of Oulu, FI-90014 Oulu, Finland. [10] Medical Research Center, Oulu University Hospital, FI-90014 Oulu, Finland. [11] Department of Clinical Genetics, Oulu University Hospital, 90220 Oulu, Finland. [12] Department of Medical Epidemiology and Biostatistics, Karolinska Institutet, SE-171 77 Stockholm, Sweden. [13] Department of Psychiatry, McLean Hospital, Belmont, MA 02478, USA. [14] Umea University, Faculty of Medicine, Department of Clinical Sciences, Psychiatry, 901 85 Umea, Sweden. [15] Psychiatry Dept., Massachusetts General Hospital, Boston, MA 02114, USA. [16] Novartis Institutes for Biomedical Research, Novartis, Cambridge, MA 02139, USA. [17] Department of Neurology, Massachusetts General Hospital, Boston, MA 02114, USA. [18] BioMarin Pharmaceutical, San Rafael, CA 94901, USA. [19]These authors contributed equally: Ralda Nehme, Olli Pietiläinen. [20]These authors jointly supervised this work: Ralda Nehme, Olli Pietiläinen, Kevin Eggan. ✉email: rnehme@broadinstitute.org; ollip@broadinstitute.org; kevin.eggan@bmrn.com

Heterozygous deletions of the 22q11.2 chromosomal interval occur approximately once in every 4000 live births[1]. The 22q11.2 deletion (22q11.2del) confers a risk of developing diverse neuropsychiatric conditions including intellectual disability (ID), Autism Spectrum Disorder (ASD) and schizophrenia[2–7]. In fact, 22q11.2del confers the largest effect of any known genetic risk factor for schizophrenia[8].

Unlike the 22q13.3 deletion syndrome, where risk of mental illness can largely be explained by the reduced function of a single gene (*SHANK3*)[9], variants in no one gene within the 22q11.2del can explain the predisposition it confers for psychiatric disease. As a result, the pathways through which it contributes to ASD and schizophrenia risk remain poorly understood. Mouse models have served as an initial inroad for identifying genes within the deletion that function in brain development and behavior. Overall, these studies suggest that several genes in the syntenic chromosomal interval including *Dgcr8*, *Ranbp1, Rtn4r*, and *Zdhhc8* have important nervous system functions[10–21]. However, imperfect alignment between mouse behavioral phenotypes and psychiatric symptoms have left uncertainty concerning which, or how many of their human orthologs play a role in mental illness.

More recent studies now suggest that the genetic background of 22q11.2del carriers contributes meaningfully to their likelihood of developing one psychiatric condition or another. For instance, deletion carriers that also harbor an additional copy number variant (CNV) elsewhere in the genome displayed a higher risk of developing schizophrenia[22]. Additionally, analysis of polygenic risk scores calculated using data from genome-wide association studies (GWAS) suggests that an increased burden of common risk variants can act in concert with 22q11.2del to further increase overall risk for psychosis[23–25]. These observations clearly indicate 22q11.2del can at least act together with alterations in genetic pathways affected by additional risk variants. This raises the possibility that the deletion may converge on disease mechanisms that act in both ASD and schizophrenia.

Finding the points of convergence between the effects of 22q11.2del and other genetic variants implicated in psychiatric disorders could thus provide a view into which genes present in the deletion, or pathways altered by it, contribute to mental illness. To identify such intersections, we examined transcriptional changes in multiple stages of excitatory neuronal differentiation, given that genetic studies of ASD and schizophrenia have implicated genes that act during neuronal development and differentiation[26–29], and in neuronal processes including excitatory transmission[30–32]. We carried out RNA sequencing at three distinct stages of excitatory neuronal differentiation using induced pluripotent stem cells (iPSCs) from 22q11.2del carriers and non-carrier controls. To establish a causal link between 22q11.2del and the transcriptional effects we also utilized gene editing to delete the chromosomal region in a control cell line. We induced neuronal differentiation using a highly reproducible approach we previously described where Ngn2 expression[33] is coupled with forebrain patterning to produce homogenous populations of excitatory neurons with features similar to those found in the superficial layers of the early cortex[34].

Here, we show that over the course of excitatory neuronal differentiation, the 22q11.2del acts in trans to significantly alter the expression of many genes with established genetic associations with neurodevelopmental disorders in progenitors, and schizophrenia in differentiated neurons. To ask, in an unbiased manner, which pathways and genes were likely responsible for these changes, we developed an approach for identifying protein-protein interaction (PPI) networks that best explain a particular change in gene expression. This method, called PPItools, suggests that 22q11.2 regulates the expression of genes in proliferative, presynaptic, proteasomal and JUN/FOS pathways. Finally, we find that cell lines with isogenic 22q11.2del recapitulate most of the changes observed in the patient-based cohort, including increased levels of the *MEF2C* transcription factor in neuronal progenitor cells and decreased expression of presynaptic proteins such as SV2A and NRXN1 in neurons.

## Results

**Pilot study and power calculations**. To study the effects of 22q11.2del, we both collected and derived hiPSC lines from patient carriers as well as non-carrier controls (Fig. 1a-f, Supplementary Fig. 1a and Table 1).

To estimate the sample size needed to detect gene expression changes, we performed a pilot study with two control and two 22q11.2del iPSC lines, each from a distinct donor. We reasoned this would allow us to detect the 50% reduction in the abundance of transcripts originating from within the deletion as well as changes in expression of genes outside of the deletion that were of a similar magnitude. We induced neuronal differentiation by combining the overexpression of Ngn2 with small-molecule patterning[35] (Fig. 1g), and completed RNA-sequencing at three cellular stages: human pluripotent stem cells (hPSCs, day 0 of differentiation), neuronal progenitor-like cells (NPCs, day 4)[35], and in functional excitatory neurons displaying synaptic connectivity[34] (day 28) (Table 1, Supplementary Fig. 1b-d). Following RNA-sequencing, we mapped reads to the Ensembl human genome assembly (GRCh37/hg19) and detected one or more reads for 51 protein-coding genes that mapped to the 22q11.2 at any one differentiation stage. We observed a systematic reduction in the abundances of RNAs encoded by these genes, with the majority exhibiting fold-changes between −1.5 and −2 in 22q11.2del cells relative to controls. Although none of the individual genes were significant after multiple testing (Supplementary Fig. 1b-d), when we considered reads from the genes in 22q11.2del in aggregate we could observe a statistically significant reduction in coding gene expression in deletion carriers (p(hPSCs) = $1.18 \times 10^{-15}$, p(NPCs) = $1.31 \times 10^{-16}$, and p(neurons) = $2.9 \times 10^{-12}$, Mann-Whitney test).

Using our pilot sequencing data, we estimated that for genes expressed above the median, a sample size of >20 carrier and >20 control iPSC lines would yield on average >80% power to detect fold-changes of 1.35 across each of the three cell stages (Fig. 1h, Supplementary Fig. 1e, f).

**Profiling an expanded 22q11.2 deletion cohort**. Guided by our power calculations, we assembled a collection of 20 (7 female, 13 male) 22q11.2del carrier and 29 (14 female, 15 male) control iPSC lines, each derived from a distinct individual. We confirmed the presence of the canonical 1.5–3 Mb deletion in 19 of the patient lines using SNP array marker intensity data. One patient line (SCBB-1430) was found to carry a smaller nested 134 kb deletion (Supplementary Fig. 1a, Table 1). While the deletion size does not seem to be correlated with diagnosis or severity of the conditions[3,7,36,37], one study found that shorter deletions might be correlated with a milder effect on IQ[38]. For this reason, while we processed this cell line similarly to the other 48, we excluded it from the main analysis in this study.

We performed RNA sequencing in hPSCs, NPCs and excitatory neurons for each of the 48 cell lines (in triplicates, N = 438 total RNA sequencing libraries in mixed pools of both genotypes to minimize technical biases). With these data in hand, we revisited our initial power estimates and found that in the larger data set we achieved over 80% power to detect fold changes ≥1.5 of most detected protein coding genes across developmental stages (Supplementary Fig. 1h).

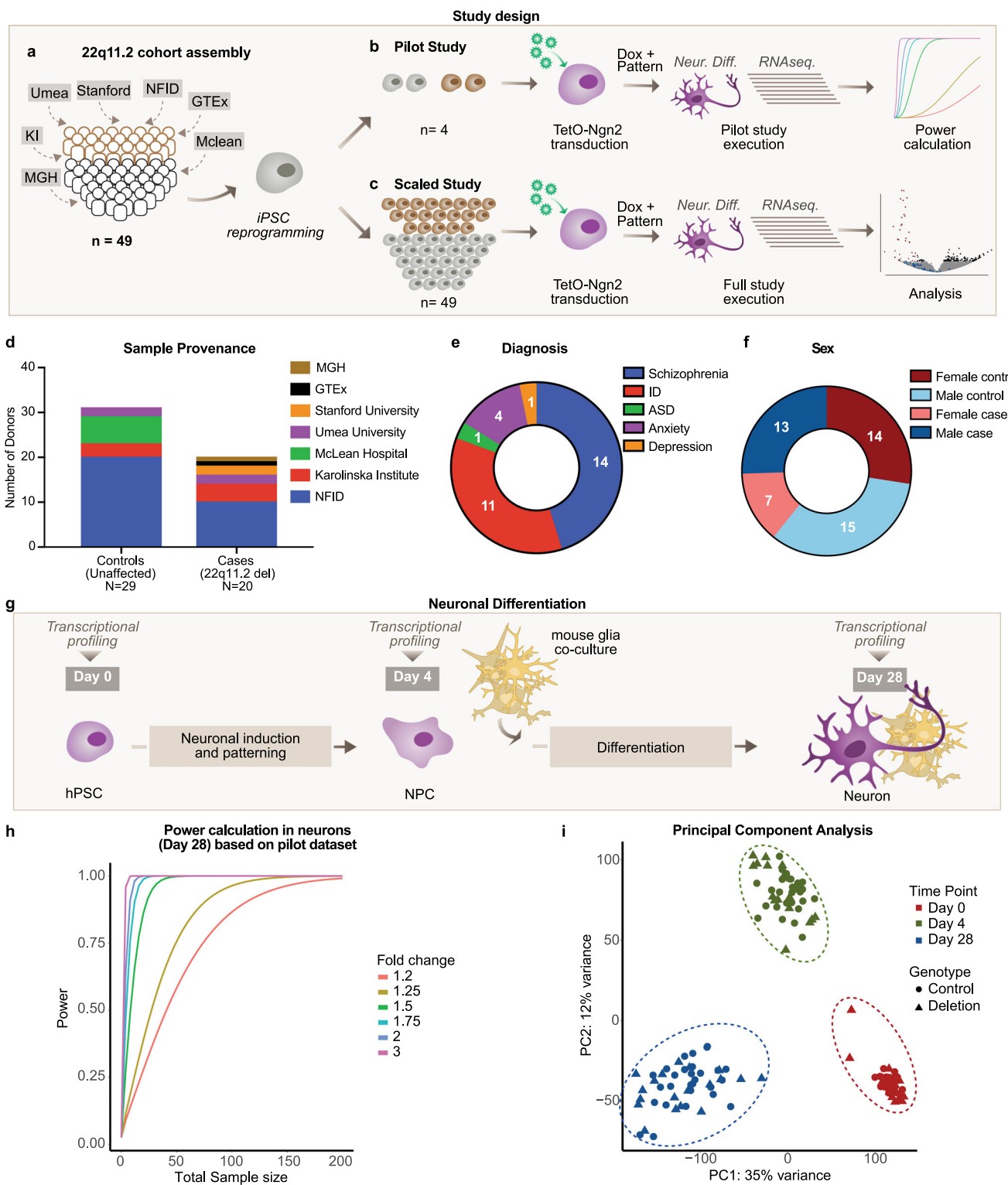

Consistent with previous findings[34,35], differentiation down a neuronal trajectory resulted in a global change of gene expression. Principal component analysis (PCA) indicated that the primary component of variation between the samples was days of neuronal differentiation (PC1 + 2 = 47% of variance) (Fig. 1i, Supplementary Fig. 1g). Close clustering of samples from the 48 lines within a given differentiation time point within PC1 and PC2 suggested reproducible differentiation across the entirety of our experiments (Fig. 1g). (This was also true for SCBB-1430,

carrying the shorter deletion). Across the 48 cell lines, 4 pluripotency-associated genes were robustly expressed at day 0 and then rapidly silenced, while 7 representative NPC genes became expressed at day 4 with the strong emergence of 7 prototypical neuronal genes at day 28 (Fig. 2a).

**22q11.2 deletion effects on transcript abundance**. We next asked how 22q11.2del influenced gene expression during neuronal differentiation and first considered genes within the deletion.

**Fig. 1 Design of a statistically powered study to determine the impact of 22q11.2 deletion on gene expression. a** Final sample set composed of 20 cell lines with 22q11.2 deletion (brown) and 29 controls (grey), collected at seven locations (MGH: Massachusetts General Hospital, KI: Karolinska Institute, Umeå: Umeå University, NFID: Northern Finnish Intellectual Disability Cohort (Institute for Molecular Medicine Finland), GTEx: Genotype-Tissue Expression Project, Mclean: Mclean Hospital). **b** Pilot study using four hiPSC lines differentiated into neurons through transduction with TetO-Ngn2, Ub-rtTA and TetO-GFP lentivirus and subjected to RNA sequencing. RNA abundances were then used to estimate the appropriate sample size for differential gene expression for the final study. **c** The final dataset consisted of 49 cell lines that were differentiated and subjected to RNA sequencing. **d** Provenance. **e** Diagnosis and (**f**), Sex of the samples in the final cohort. **g** Neuronal differentiation protocol (Nehme et al 2018) consisting of the combination of Ngn2 overexpression with forebrain patterning using small molecules (SB431542, LDN193189 and XAV939). Samples were harvested for RNA sequencing at the stem cell (day 0), neuronal progenitor cell (NPCs) (day 4) and neuronal (day 28) stages. **h** Power estimation in the pilot dataset for median expressed genes (24 read counts) for different fold-changes and sample sizes in neurons. **i** Principal component analysis (PCA) of RNA sequencing data from the full study.

We observed a nominally significant reduction in RNA levels for 59 genes within the deletion ($p < 0.05$, red and blue dots, Fig. 2c-e) with 48 significantly reduced in at least one time point (FDR < 0.05) and 27 significantly reduced in all 3 stages (FDR < 0.05) (Fig. 2b, Supplementary Fig. 2b). These findings in excitatory neuronal cells were in line with previous reports using either mixed cultures of inhibitory and excitatory neurons[39] or organoids consisting of multiple cell types including glutamatergic neurons and astrocytes[40]. For significantly altered genes mapping to the deletion, the deletion genotype explained, on average, 45–54% of all variance in their expression (Supplementary Fig. 3a-d). These 48 genes included seven that are highly intolerant for loss of function variants as measured by pLI score[41], which ranks genes from most tolerant (pLI = 0) to most intolerant (pLI = 1). These genes (pLI score >0.9; *UF1DL, HIRA, DGCR8, ZDHHC8, MED15, TBX1, SEPT5*) have been previously suggested to play role in some of the congenital phenotypes associated with 22q11.2del in other tissues[42]. Together, our analyses indicate that our transcriptional phenotyping was sufficiently sensitive to allow the successful detection of the 50% decrease in expression of the hemizygote genes found in the deletion region.

**Cell-type-specific effects of 22q11.2 deletion**. We next explored differentially expressed transcripts originating from loci outside 22q11.2del. In fact, the majority (89%) of the genes differentially expressed in 22q11.2del cells were located outside the deletion ($n = 386$ genes) (Fig. 2b). In total, such trans effects explained on average 18% of the total variance in gene expression across all data sets (Supplementary Fig. 3a-d). Plotting the test statistic from the differential expression for every gene relative to its position in the genome suggested that there was no major positional clustering of differentially regulated genes to specific chromosomal regions outside chromosome 22 (Supplementary Fig. 3e, day 28 example). Only one gene, *CAB39L* and a pseudogene, *TPTEP1* on chromosomes 13 and 22, were significantly induced in carriers at all stages (Supplementary Fig. 2c). Notably, *CAB39L* expression was also induced in blood cells isolated from 22q11.2del carriers[43], suggesting that upregulation of this gene is likely to be associated with the 22q11.2del in many cell types.

While genes within 22q11.2del were regulated in the same direction at all developmental stages, except for *CAB39L* and *TPTEP1*, the set of differentially expressed genes outside the region was different for each stage (383 cell stage-specific genes). Importantly, in controls, the affected genes were expressed in all cell stages with little change between stages (Supplementary Data 1–4 and Supplementary Fig. 4a, b). As a result, fold-changes in "trans" genes between carriers and controls were only modestly correlated between NPCs and hPSCs ($\rho = 0.35$, $p = 3 \times 10^{-11}$), NPCs and neurons ($\rho = 0.25$, $p = 3.4 \times 10^{-6}$). and hPSCs and neurons ($\rho = 0.12$, $p = 0.02$). Differential gene expression analysis in day 28 neurons datasets either including or excluding the cell

line with the small deletion (SCBB-1430) was strongly correlated ($r = 0.99$ for genes with adjusted $p$ value < 0.05), suggesting that the observed gene expression differences were robust also in the presence of the shorter deletion. The resulting 2% increase in sample size upon inclusion of SCBB 1430 yielded a slightly higher number of differentially expressed genes (432).

These findings suggest that 22q11.2del has a temporally dependent influence on gene expression, altering the abundance of distinct sets of transcripts as neuronal differentiation unfolds.

**Transcript alterations in hPSCs and NPCs**. The phenotypes that are found in a subset of 22q11.2del carriers during early childhood[3] led us to ask if the genes differentially expressed at initial differentiation stages (hPSCs and NPCs) were genetically associated with neurodevelopmental disorders. We included likely disease-causing genes from the Deciphering Developmental Delay (DDD) project, and a recent, large exome-sequencing study in autism ($n = 295$ total neurodevelopmental disorders, NDD, genes)[28,44,45] (Supplementary Data 5). Of the 434 genes we found differentially expressed in deletion carriers, 10 were NDD genes (hPSCs: *ELAVL3*; NPCs: *PAX6, MEF2C, FOXP2, NR2F1, PAX5, TBX1*; neurons: *KMT2C, MKX*; OR = 1.6, $p = 0.1$ (for all 434 genes) (Supplementary Data 1–3). Notably, *MEF2C* is also implicated in schizophrenia through GWAS[46] and encodes a transcriptional regulator of activity-dependent immediate early genes such as *JUN* and *FOS*[47]. *MEF2C* has been shown to be repressed by the transcription factor *TBX1*, encoded by a gene within 22q11.2[48,49].

Proteins encoded by genes harboring causal variants for a particular phenotype in Mendelian disorders have been shown to have more physical connections between one another than unrelated proteins[50]. We therefore wondered whether the transcripts expressed from within 22q11.2del and the transcripts with altered abundance in trans encoded proteins that together had more than the expected number of interactions with proteins originating from loci genetically linked with NDD. As this is a question of broader relevance for connecting protein interaction data, changes in gene expression, and genetic data, we wrote a software package[51] to enable this analysis.

In this instance we used PPItools to identify the protein-protein interactions (PPI) from the InWeb database[52] of the differentially expressed gene products at each stage of neural differentiation and analyzed them for an apparent excess of genes implicated in NDD in this network. We used a curated list of NDD genes that comprised 295 genes that have been previously reported to have excess of deleterious variants in patients with ASD, and ID[28,44,45] (Supplementary Data 5). To ask whether this enrichment for NDD-implicated interacting proteins was likely to have occurred by chance, we performed 1000 random permutations of sets of expressed proteins of the same size while constraining the scale and complexity of the network. These analyses confirmed that genes we found to be differentially expressed early in differentiation (in hPSCs and NPCs) were

**Table 1 Description of samples in the study.**

| Lines | Gender | Age (years) | Genetics | Diagnosis | hiPSC reprogramming facility | Deletion Start | Deletion End | Deletion Length (bp) | Deletion Type | Notes |
|---|---|---|---|---|---|---|---|---|---|---|
| SCBB-1827 | Female | 60+ | Control | Control | In house | NA | NA | NA | NA | Used in Pilot study |
| SCBB-1828 | Male | 40-59 | Control | Control | In house | NA | NA | NA | NA | Used in Pilot study |
| SCBB-228 | Female | 20-39 | Control | Control | NYSCF | NA | NA | NA | NA | |
| SCBB-229 | Male | 20-39 | Control | Control | NYSCF | NA | NA | NA | NA | |
| SCBB-243 | Male | 20-39 | Control | Control | NYSCF | NA | NA | NA | NA | |
| SCBB-258 | Female | 60+ | Control | Control | NYSCF | NA | NA | NA | NA | |
| SCBB-269 | Male | 20-39 | Control | Control | NYSCF | NA | NA | NA | NA | |
| SCBB-800 | Female | 40-59 | Control | Control | Harvard iPS core | NA | NA | NA | NA | |
| SCBB-799 | Female | 20-39 | Control | Control | Harvard iPS core | NA | NA | NA | NA | |
| SCBB-798 | Male | 20-39 | Control | Control | Harvard iPS core | NA | NA | NA | NA | |
| SCBB-803 | Female | 20-39 | Control | Control | Harvard iPS core | NA | NA | NA | NA | |
| SCBB-1480 | Male | 40-59 | Control | Control | Harvard iPS core | NA | NA | NA | NA | |
| SCBB-1485 | Male | 60+ | Control | Control | Harvard iPS core | NA | NA | NA | NA | |
| SCBB-1489 | Female | 60+ | Control | Control | Harvard iPS core | NA | NA | NA | NA | |
| SCBB-1473 | Male | 40-59 | Control | Control | Harvard iPS core | NA | NA | NA | NA | |
| SCBB-1645 | Female | 40-59 | Control | Control | Harvard iPS core | NA | NA | NA | NA | |
| SCBB-1477 | Female | 20-39 | Control | Control | Harvard iPS core | NA | NA | NA | NA | |
| SCBB-1478 | Male | 20-39 | Control | Control | Harvard iPS core | NA | NA | NA | NA | |
| SCBB-1646 | Male | 40-59 | Control | Control | Harvard iPS core | NA | NA | NA | NA | |
| SCBB-1647 | Female | 20-39 | Control | Control | Harvard iPS core | NA | NA | NA | NA | |
| SCBB-1479 | Male | 60+ | Control | Control | Harvard iPS core | NA | NA | NA | NA | |
| SCBB-1648 | Female | 40-59 | Control | Control | Harvard iPS core | NA | NA | NA | NA | |
| SCBB-1471 | Female | 20-39 | Control | Control | Harvard iPS core | NA | NA | NA | NA | |
| SCBB-1472 | Male | 40-59 | Control | Control | Harvard iPS core | NA | NA | NA | NA | |
| SCBB-1483 | Male | 40-59 | Control | Control | Harvard iPS core | NA | NA | NA | NA | |
| SCBB-1481 | Female | 20-39 | Control | Control | Harvard iPS core | NA | NA | NA | NA | |
| SCBB-652 | Male | 40-59 | Control | Control | Harvard iPS core | NA | NA | NA | NA | |
| SCBB-653 | Female | 40-59 | Control | Control | Harvard iPS core | NA | NA | NA | NA | |
| SCBB-1438 | Male | 40-59 | Control | Control | Harvard iPS core | NA | NA | NA | NA | |
| SCBB-1825 | Male | 40-59 | 22q11.2del | Schizophrenia | in house | 18655798 | 21726191 | 3070393 | LCR22A-D | Used in Pilot study |
| SCBB-1445 | Male | 40-59 | 22q11.2del | Schizophrenia | NYSCF | 18650803 | 21800227 | 3149424 | LCR22A-D | |
| SCBB-1962 | Female | 40-59 | 22q11.2del | Schizophrenia | in house | 18875445 | 21461607 | 2586162 | LCR22A-D | Used in Pilot study |
| SCBB-1530 | Female | 10-19 | 22q11.2del | Mild ID, anxiety, psychosis | NYSCF | 18650726 | 21800227 | 3149501 | LCR22A-D | |
| SCBB-1961 | Male | 2-9 | 22q11.2del | Autism Spectrum Disorder, ID | Stanford | 19470249 | 21028007 | 1557758 | LCR22A-B | |
| SCBB-1960 | Male | 20-39 | 22q11.2del | Schizophrenia, Anxiety | Stanford | 18875445 | 21563155 | 2687710 | LCR22A-D | |
| SCBB-1430 | Male | 60+ | 22q11.2del | Schizophrenia | NYSCF | 18892575 | 19026633 | 134058 | NA | |
| SCBB-1524 | Male | 40-59 | 22q11.2del | Schizophrenia | NYSCF | 18650803 | 21799719 | 3148916 | LCR22A-D | |
| SCBB-1434 | Male | 40-59 | 22q11.2del | Schizophrenia | NYSCF | 18650803 | 21799719 | 3148916 | LCR22A-D | |
| SCBB-1437 | Female | 20-39 | 22q11.2del | Mild ID | NYSCF | 18797591 | 21809133 | 3011542 | LCR22A-D | |
| SCBB-290 | Male | 60+ | 22q11.2del | Moderate ID, Schizophrenia | Harvard iPS core | 18650726 | 21800227 | 3149501 | LCR22A-D | |
| SCBB-963 | Male | 40-59 | 22q11.2del | Schizophrenia | Harvard iPS core | 18637094 | 21726191 | 3089097 | LCR22A-D | |
| SCBB-937 | Female | 20-39 | 22q11.2del | Mild ID, Anxiety | Harvard iPS core | 18889969 | 21563155 | 2673186 | LCR22A-D | |
| SCBB-801 | Male | 2-9 | 22q11.2del | Mild ID | Harvard iPS core | 18655798 | 21804903 | 3149105 | LCR22A-D | |
| SCBB-1652 | Female | 40-59 | 22q11.2del | Schizophrenia | Harvard iPS core | 18650746 | 21799719 | 3148973 | LCR22A-D | |

**Table 1 (continued)**

| Lines | Gender | Age (years) | Genetics | Diagnosis | hiPSC reprogramming facility | Deletion Start | Deletion End | Deletion Length (bp) | Deletion Type | Notes |
|---|---|---|---|---|---|---|---|---|---|---|
| SCBB-1499 | Male | 20–39 | 22q11.2del | Moderate ID, Schizophrenia | Harvard iPS core | 18650746 | 21799719 | 3148973 | LCR22A-D | |
| SCBB-1500 | Female | 40–59 | 22q11.2del | Moderate-Severe ID, Anxiety, Psychosis | Harvard iPS core | 18650803 | 21799719 | 3148916 | LCR22A-D | |
| SCBB-1503 | Male | 20–39 | 22q11.2del | Moderate ID, Schizophrenia | Harvard iPS core | 18650746 | 21799719 | 3148973 | LCR22A-D | |
| SCBB-1504 | Male | 20–39 | 22q11.2del | Moderate ID, Depression | Harvard iPS core | 18801657 | 21799890 | 2998233 | LCR22A-D | |
| SCBB-1490 | Female | 10–19 | 22q11.2del | Mild ID | Harvard iPS core | 18650746 | 21799719 | 3148973 | LCR22A-D | |
| H1-A | Male | NA | Control | NA | NA (hESC) | NA | NA (hESC) | NA | NA | isogenic lines |
| H1-D | Male | NA | Control | NA | NA (hESC) | NA | NA (hESC) | NA | NA | isogenic lines |
| H1-E | Male | NA | 22q11.2del | NA | NA (hESC) | 18924956 | 21460220 | 2535264 | LCR22A-D | isogenic lines |
| H1-G | Male | NA | 22q11.2del | NA | NA (hESC) | 19175095 | 21460220 | 2285125 | LCR22A-D | isogenic lines |

Coordinates on Chromosome 22 are indicated to delineate the deletion start (column H) and end (column I) estimated based of SNP array data, using Human Genome build 19.

significantly more likely to interact with gene products associated with NDD ($p < 0.001$, Supplementary Fig. 4c). This enrichment was not significant in neurons.

To further control our observation, we asked whether the protein interaction network we identified at each time point showed any enrichment for genes linked with an unrelated condition, inflammatory bowel disease (IBD), or with a non-neurodevelopmental neurological condition, Parkinson's Disease (PD). As expected, there were no significant enrichments for IBD-related gene products within the networks identified at any of the stages analyzed, and no enrichment for PD-related gene products in NPCs or neurons (Supplementary Fig. 4c, Supplementary Data 5). Thus, our results demonstrate that within hPSCs and NPCs, there is indeed a convergence between genes within 22q11.2del and the transcripts altered in trans with genes products linked to human neurodevelopmental disorders.

**Schizophrenia heritability enrichment in neurons.** Given that we had found an initial convergence between the effects of 22q11.2del and the abundance of certain transcripts linked to NDD, we next asked whether the transcripts that were altered in 22q11.2del cells were enriched for additional genetic signals in mental illness. We utilized the genes with significantly altered expression as a substrate for linkage disequilibrium (LD)-score regression[30]. For this analysis we used GWAS summary statistics from the psychiatric genomics consortium (PGC), as well as educational attainment studies[53–58] to ask whether variants in 22q11.2-differentially expressed genes and their surrounding genomic regions contribute disproportionately to the polygenic heritability of five neuropsychiatric disorders (schizophrenia, bipolar disorder, major depressive disorder, autism spectrum disorder, and ADHD). The analysis with LD-score regression revealed an enrichment in heritability for schizophrenia among genes differentially expressed in neurons (1.20-fold enrichment, $\tau_c = 1.0 \times 10^{-8}$; $p$- adjusted (Bonferroni) $= 0.01$ for all 3,370 genes with nominally significant differences in expression, $p < 0.05$, including 2173 up genes and 1,197 down genes, respectively) (Fig. 3a). Analysis of up- and downregulated genes separately revealed that the increase in the heritability was accounted for by transcripts that were more abundant in 22q11.2del neurons (1.31-fold enrichment, $\tau_c = 1.96 \times 10^{-8}$, $p$-adjusted 0.001) (Fig. 3a, Supplementary Data 6). Our findings were unlikely to be the result of neurons expressing increased levels of genes relevant for these psychiatric conditions: permutation with 100 random gene lists produced from our neuronal data and matched for expression level indicated that the observed heritability enrichment was greater than any random gene set ($p$ (empirical) $<0.01$) (Supplementary Fig. 5a-c).

To confirm our results using an independent method, we queried the relationship between differentially expressed genes in 22q11.2del neurons and common genetic variants associated with psychiatric illness with a different set of statistical assumptions. We applied multiple-regression for competitive gene-set analysis in MAGMA-software[59]. Like results from the LD-score regression analysis, genes whose transcripts were more abundant in 22q11.2del neurons were more strongly associated with schizophrenia than the rest of the genome ($p = 7.93 \times 10^{-8}$, p(Bonferroni)$=5.07 \times 10^{-6}$) (Supplementary Fig. 6a, Supplementary Data 7). Altogether, 13 genes with nominally significant gene-wise association to schizophrenia from MAGMA ($p_g < 0.05$) were significantly differentially expressed in deletion neurons (Fig. 3b, Supplementary Fig. 6c). Repeating the analysis with 100 random gene lists generated from our expression data confirmed that this result was unlikely to have arisen merely as a result of examining these neuronal cells (Supplementary Fig. 6b).

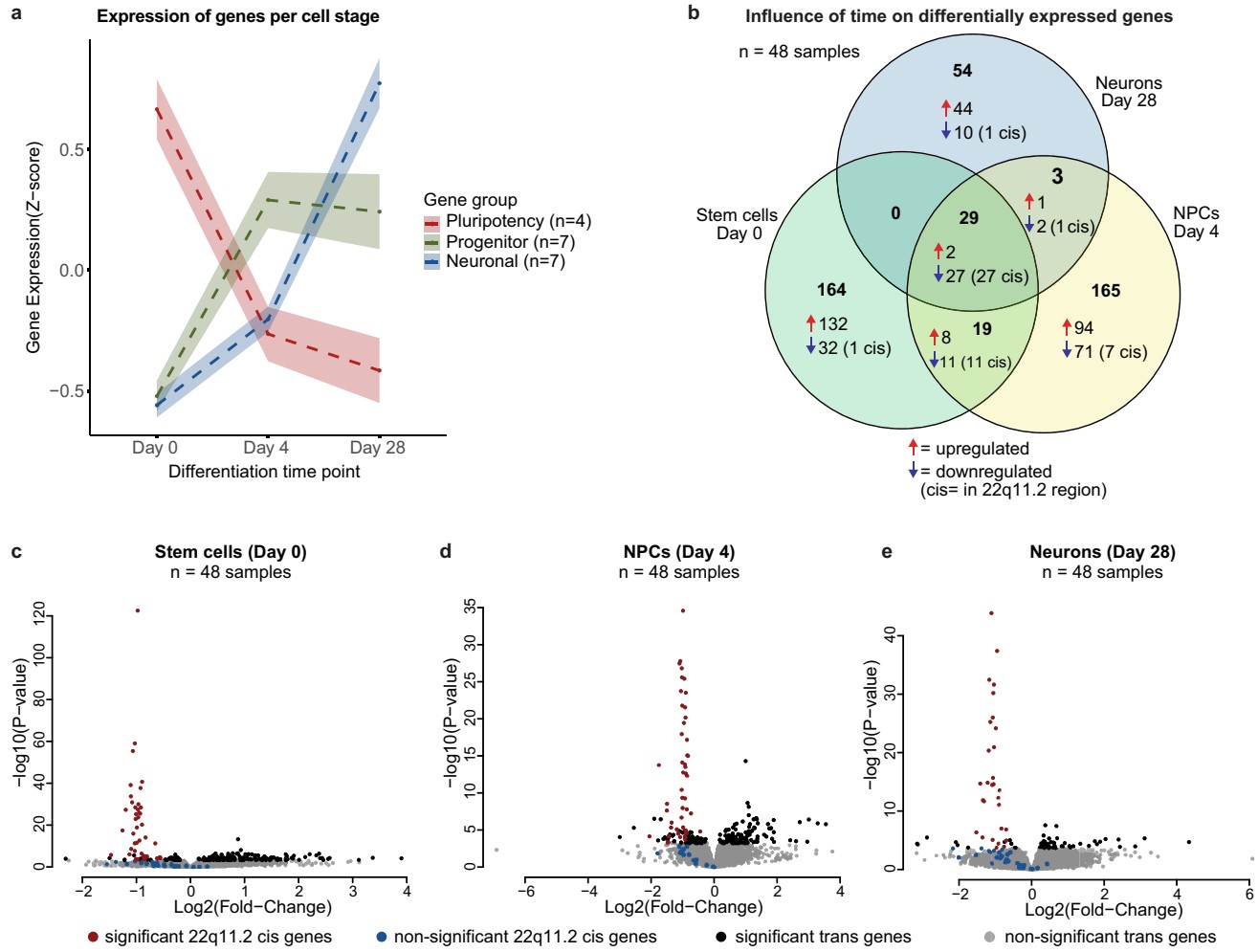

**Fig. 2 Cell-type-specific effects of the 22q11.2 deletion. a** Expression of selected marker genes and 95%-confidence interval of the mean for defined specific cell stages by suppression of genes related to pluripotency (*SOX2, OCT4, NANOG, MKI67*) and up-regulation of genes characteristic for neural progenitor cells (*NEUROD1, SOX2, EMX2, OTX2, HES1, MSI1, MKI67*) and mature neurons (*NEUN, SYN1, DCX, MAP2, TUJ1, NCAM, MAPT*) as the differentiation progresses. **b** Venn Diagram highlighting the number and directionality of shared and unique differentially expressed genes (FDR < 5%) between deletion carriers and controls at each cell stage. Genes within the deletion region (cis) are mostly shared across development stages, whereas genes outside the deletion region (trans) are cell-stage specific. **c–e**, Volcano plots showing differential gene expression in stem cells (**c**), NPCs (**d**) and neurons (**e**) Significantly differentially expressed genes (FDR < 5%) within the deletion region are presented in red and outside deletion in black. Non-significant genes in deletion region are presented in blue. The test statistics were derived from Wald-test in DEseq2 and are presented in Supplementary Data 1–3.

Notably, when we included the cell line with the short deletion (SCBB 1430) in the analysis, the enrichment for schizophrenia heritability remained similar in both LD-score regression (1.50-fold enrichment, $p = 1.05 \times 10^{-13}$) and MAGMA ($p = 5.6 \times 10^{-7}$). To determine if this association between 22q11.2del induced genes and schizophrenia heritability was replicable and specific, we used summary statistics from an independent GWAS dataset of 650 heritable traits from the UK-biobank. LD-score regression showed the genes upregulated in 22q11.2del neurons harbored significant heritability enrichment for schizophrenia, but not for the other traits (Supplementary Fig. 6d). Overall, our findings indicated that excitatory neurons harboring the 22q11.2del exhibited increased abundance of transcripts from genes that underlie schizophrenia heritability, but that the deletion did not have such a detectable effect at earlier stages of differentiation.

**Schizophrenia rare variant enrichment in neurons**. Exome sequencing at increasing scale has begun to reveal a burden of rare protein damaging variants in schizophrenia patients, complementing the genetic signal of common regulatory variants emerging from GWAS[60–62]. In contrast to the common variant polygenic risk, which arises incrementally from many small-effect variants, the schizophrenia-associated rare variants identified so far act with strong individual effects. While there is evidence for common and rare risk variants in schizophrenia mapping to shared chromosomal intervals[32], so far the two forms of variation implicate partially distinct sets of genes. We therefore asked whether 22q11.2del also effects the expression of genes that harbor rare coding variants, identified by the schizophrenia exome meta-analysis consortium (SCHEMA) in schizophrenia patients[62–64]. We initially focused on genes upregulated in 22q11.2del neurons ($n = 2,173$ genes at $p < 0.05$) and used 100 random gene lists matched for their expression levels in our excitatory neurons as controls. This analysis revealed two interesting results: First, many of the genes abundantly expressed in neurons harbor a burden of loss of function mutations in schizophrenia (Fig. 3d red dots, 51/100 random gene lists assessed $p < 0.05$). Second, the 2,173 transcripts within these neurons

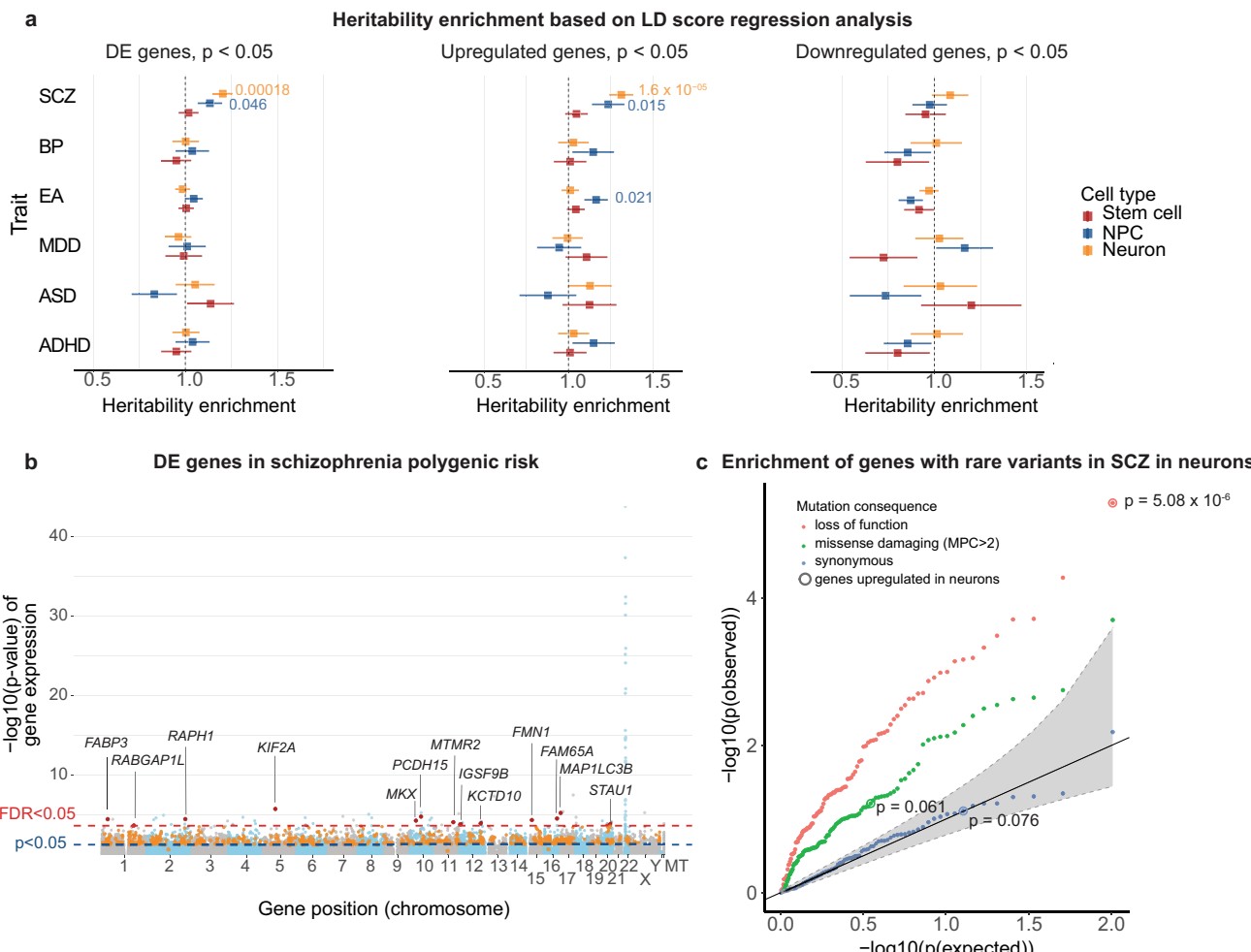

**Fig. 3 Heritability enrichment for schizophrenia risk genes 22q11.2 deletion neurons. a** Marginal enrichment in Heritability ($+/-$ SE) explained by common (MAF > 5%) variants within 100 kb of genes differentially expressed, estimated by LD Score regression. Six traits were analyzed: SCZ schizophrenia, BP bipolar disorder, EA educational attainment, MDD major depressive disorder, ASD autism spectrum disorder, ADHD attention deficit hyperactivity disorder, at all three cell stages, showing enrichment for schizophrenia most prominently in genes upregulated in 22q11.2 deletion neurons. DE differentially expressed. Three groups of DE genes were analyzed. Left, all nominally significant DE genes ($p < 0.05$, N (DE genes in stem cells) = 2346, N (DE genes in NPCs) = 2076, N (DE genes in neurons) = 3370). Middle, all nominally significant upregulated DE genes ($p < 0.05$, N (genes in stem cells) =1521, N (genes in NPCs) = 964, N (genes in neurons) =2173). Right, all nominally significant downregulated DE genes ($p < 0.05$, N (genes in stem cells) =825, N (genes in NPCs) = 1112, N (genes in neurons) =1197). The standard errors in LD score regression are derived from block Jackknife that are then used to calculate z scores and $P$ values (two-sided). A significant heritability enrichment for schizophrenia was found for all DEgenes and upregulated genes in neurons ($p = 0.00018$ and $p = 1.6 \times 10^{-5}$, LD score Regression). The test statistic for each comparison is presented in Supplementary Data 6. **b** DE genes (two-sided Wald-test implemented in DEseq2) in 22q11.2 neurons with nominally significant gene-wise association to schizophrenia from MAGMA ($p_g < 0.05$, F-test implemented in MAGMA, two-sided). **c** qq plot of p-values (t-test, linear regression, two-sided) with 95% confidence interval for the enrichment of rare coding LoF, missense damaging or synonymous variants in schizophrenia patients in genes upregulated in deletion neurons (circled) and 100 random gene sets matched by expression level to the upregulated genes.

whose expression were increased in 22q11.2del carriers were more significantly enriched for loss of function variants than any of the random gene lists we sampled (Fig. 3d, red dot black circle; $p = 5.08 \times 10^{-6}$). This enrichment signal was reduced for missense mutations in schizophrenia patients and absent for synonymous variants (green and blue dots Fig. 3d). We note that none of the SCHEMA genes had significantly (FDR < 5%) different transcript abundances in the deletion lines compared to controls.

Consistent with the notion that we were analyzing a disease-relevant cell type, our rare-variant burden analyses indicated that the excitatory neurons we produced from both cases and controls expressed a significant excess of genes harboring rare pathogenic coding variants in schizophrenia patients. However,

our analysis further indicated that in excitatory neurons the 22q11.2del was specifically associated with alterations in a set of genes that were even more markedly enriched for rare loss of function variants in schizophrenia patients (Fig. 3d, circled dot). Like our common variants analyses, genes whose expression we found altered in 22q11.2del hPSCs and NPCs did not exhibit this excess of rare coding variants (Supplementary Fig. 7a, Supplementary Data 8).

**Protein-protein interaction networks associated with transcriptional changes.** As the number of trans-acting effects of the deletion on transcripts linked to psychiatric illness were substantial, we sought an unbiased approach for identifying the

pathways that could be contributing to their alterations. Ideally, such a method would identify potential connections to gene products originating from within the deletion interval. To this end, we used PPI data[52] to search for the smallest number of biochemical interactions that could explain the most prominent transcriptional changes in 22q11.2del carriers. We developed a new tool (included in the R-package "PPItools", see methods) that scores observed p-values from differential expression to construct a node-weighted graph with the strongest cumulative association with the deletion genotype at each cell stage (most-weighted connected subgraph, MWCS). MWCS provides an approach to identify a set of genes with the strongest collective association in PPI network without a predefined significance threshold, by modeling the p-value distribution of the differentially expressed genes and combining this with network permutations that maintain the key PPI network characteristics.

We then performed 1000 permutations on p-values from differential expression while preserving the node degrees, to ensure that the connected gene products were unlikely to occur in the subgraph by chance alone ($p < 0.05$, Supplementary Data 9, Supplementary Fig. 8a). This analysis revealed that the minimal interaction networks for each differentiation stage were predominantly composed of proteins encoded by 22q11.2 genes, that were in turn interconnected with proteins encoded by genes outside the region (Supplementary Fig. 8b, c and Fig. 4g).

In hPSCs, we found that the MWCS contained 45 node proteins, 25 of which were encoded by genes mapping to 22q11.2del (Supplementary Fig. 8b). These nodes were organized around several hub proteins encoded by genes that map outside the deletion. These included MYC, p53 (TP53) and the autism-associated protein p21 (CDKN1A) suggesting that the deletion disrupts regulation of the cell cycle and directly impacts expression. Our analyses suggest alterations in the expression of these well-known cell cycle regulators could be mediated by reduced expression of several interacting proteins that map to 22q11.2del including CDC45, a regulator of DNA replication, TRMT2A, which encodes a known cell cycle inhibitor, as well as LZTR1 a known tumor suppressor. Another hub was that encoding the low-affinity nerve growth factor receptor NGFR and known NOGO Co-receptor P75, along with NOGO receptor (RTN4R) and the mediator of protein degradation through the proteasome UFD1L, both encoded within the deletion.

In NPCs (Supplementary Fig. 8c), we continued to see evidence for disruption in NOGO signaling through increased expression of both NOGO (RTN4) and the TRKA receptor (NTRK1), which is associated with autism through rare protein-coding variation, is a known interactor with P75 and is modulated by NOGO signaling. These findings suggest that reduced expression 22q11.2del proteins such as the NOGO receptor and less appreciated interacting proteins encoded within 22q11.2del such as PIK4A and ARVCF are disrupting signaling. Another signal from the minimal network in NPCs was for a disruption in RNA metabolism. This was exemplified by a hub centered around the TFIID transcription factor, TAF1 which interacted with the tumor suppressor proteins LZTR1 and LZTS2, MOV10, and GNB1L, encoded within the deletion, with roles in cell cycle progression and gene regulation.

In neurons (Fig. 4g), we identified three major hubs consisting of 1) interactors of the activity-dependent transcription factor *JUN*, including the proteasome subunit PSMD12 and the kinesin KIF2A, both associated with NDD, and several proteins encoded in 22q11.2: TRMT2A, RANBP1, GNB1L, MRPL40, SCL25A1, CRKL, with connections to the transcriptional (POLR2A) and chromatin remodeling (HIRA) machineries; 2) components of the protein ubiquitination/metabolism pathway, including SMAD2, COPS5 and WWP2 along with UFD1L and KLHL22, both encoded within

the deletion region; and 3) synaptic vesicle trafficking, including CLTCL1 encoding clathrin, the synaptobrevin VAMP2, which is associated with NDD, and SNAP29 located in the 22q11.2 locus and encoding a synaptosome associated protein (Fig. 4g). Overall, our analyses support the notion that multiple distinct but connected pathways may be at the core of the transcriptional changes that we observe in 22q11.2del neurons: activity-dependent gene expression, protein homeostasis, and synaptic biology.

**Enrichment of synaptic, activity-dependent gene expression and protein homeostasis programs in deletion altered transcripts.** We next wondered how changes in gene expression caused by 22q11.2del might impact neurobiological processes. To this end, we first employed recently reported synaptic gene ontologies[31] to search for potentially converging synaptic biology among the genes differentially expressed in 22q11.2del neurons. 193 of the 2,173 transcripts with increased abundance ($p < 0.05$) in 22q11.2 neurons possessed a synaptic process annotation in SynGO[31] ($q = 1.1 \times 10^{-4}$), with a particular enrichment for transcripts with presynaptic functions in synaptic vesicle cycle (GO:0099504, $q = 0.001$, Fig. 4a, Supplementary Data 10), while 41 of the 1203 downregulated transcripts, including five cis genes, had a SynGO annotation. We next wondered whether these 193 synaptic genes were a major contributor to the schizophrenia heritability enrichment we detected. Removing the 193 synaptic transcripts from the 2173 with increased abundance in 22q11.2del neurons had only little effect on the heritability enrichment for schizophrenia, a similar effect to when randomly drawn lists of 193 transcripts were removed from the pool of 2173 transcripts (Fig. 4b).

Including the cell line with the shorter deletion enhanced the synaptic signal detected in the induced genes, likely due to the resulting increased power in our dataset (239 genes with SynGO annotations identified, Supplementary Fig. 7b, c). The additional synaptic elements from the 2% increase in sample size contributed proportionally more to the estimated enrichment in heritability of schizophrenia that decreased from 1.5 to 1.4 (stde = 0.065, $p = 1.3 \times 10^{-10}$) fold-enrichment after removing the 239 synaptic transcripts. This decrease was larger than observed for 100 randomly drawn lists of 239 genes (Supplementary Fig. 7c). A further gene ontology enrichment analysis revealed that genes induced in 22q11.2del neurons were significantly enriched for functions in the protein ubiquitination pathway (GO:0000209, 82 genes, OR = 2.67, $q = 7.8 \times 10^{-13}$) and among the largest individual enrichments was regulation of synaptic vesicle cycle (GO:0098693, 15 genes, OR = 3.2, $q = 1.96 \times 10^{-3}$) (Supplementary Data 11). This enrichment with functions in protein homeostasis and synaptic signaling was specific for induced genes in neurons. Genes induced in NPCs were enriched for skeletal system development (GO:0001501, 50 genes, OR = 2.3, $q = 0.0002$). Genes induced in stem cells did not highlight any specific biological process. In comparison, genes with reduced expression in neurons highlighted exclusively functions in cilium assembly (GO:0060271, 49 genes, FC = 2.3, $q = 1.2 \times 10^{-7}$), while genes reduced in hPSCs and NPCs were not enriched for any biological processes (Supplementary Data 12–13).

We further carried out motif enrichment analysis on the genes upregulated ($p < 0.05$) in all 22q11.2del neurons to identify transcription factor binding motifs that are enriched in this gene set. The motif that was most significantly enriched was for binding of the *JUN/FOS* transcription factors (1.6-fold enrichment, $p = 10^{-14}$; Supplementary Fig. 7d, Supplementary Data 14). The *JUN* and *FOS* transcription factors are immediate-early genes that are activated in response to neurotransmitter release and activate a downstream "activity-dependent" transcriptional

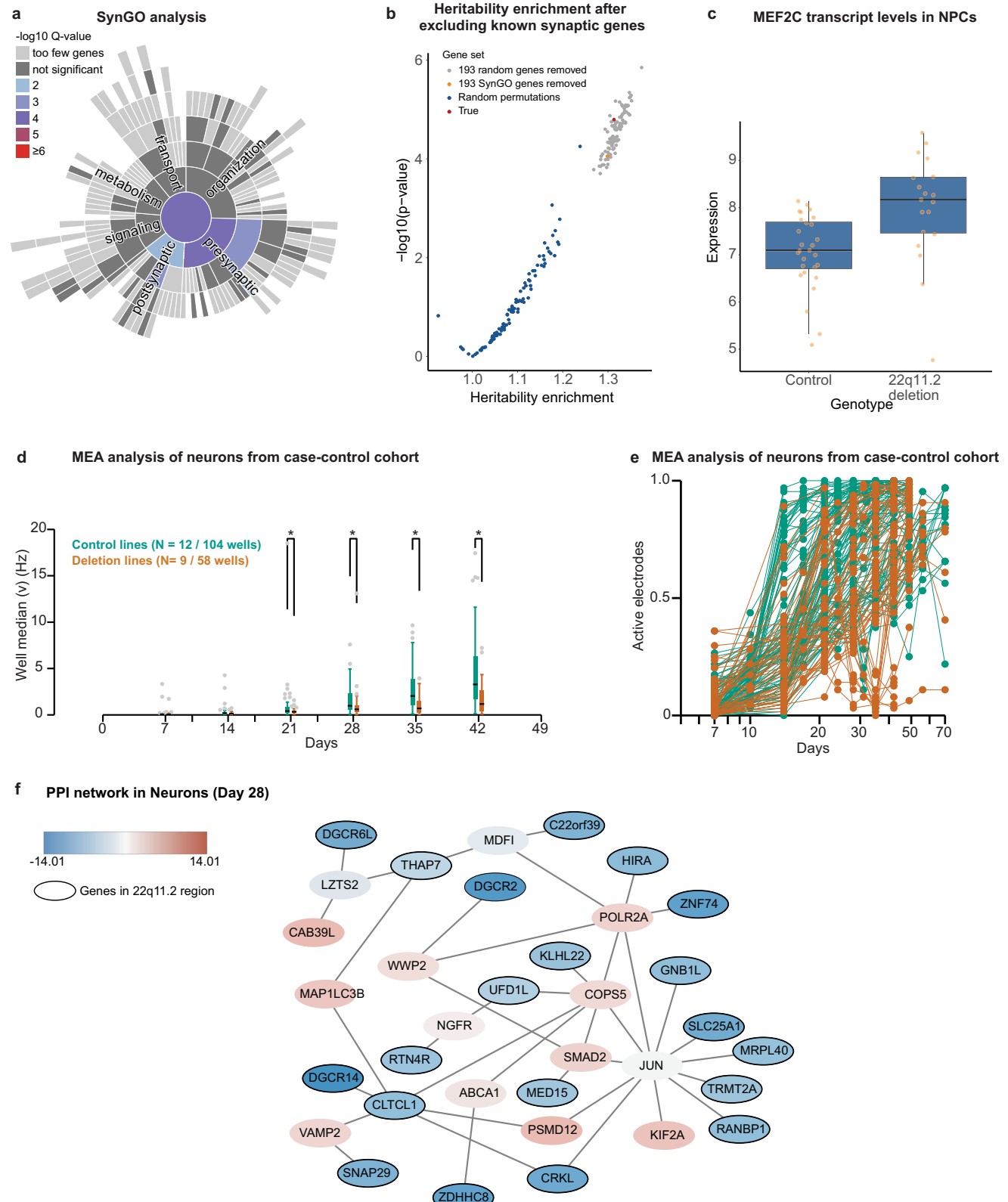

**a** SynGO analysis

**b** Heritability enrichment after excluding known synaptic genes

**c** MEF2C transcript levels in NPCs

**d** MEA analysis of neurons from case-control cohort

**e** MEA analysis of neurons from case-control cohort

**f** PPI network in Neurons (Day 28)

cascade to regulate downstream programs, such as protein homeostasis and synaptic transmission[47].

Notably, transcript levels of *MEF2C*, an activity-dependent transcription factor acting upstream of the JUN / FOS signaling pathway to regulate the expression of immediate early genes[47], are increased in 22q11.2del NPCs in our discovery dataset (Supplementary Data 2, Fig. 4c, and validated by qPCR and immunoblotting, Supplementary Fig. 2f, g, 10). *MEF2C* has been shown to negatively regulate synaptic transmission by restricting the number of excitatory synapses[65,66]. Additionally, *TBX1*, a transcription factor located in 22q11.2del and significantly down-regulated in these same NPCs (Supplementary Fig. 2d, e), is a known repressor of *MEF2C*[48,49]. Thus, decreased *TBX1* levels due to loss of a copy of 22q11.2 likely result in de-repression of the

**Fig. 4 Impact of the 22q11.2 deletion on synaptic gene expression and network activity. a** SynGO annotation for genes upregulated in neurons ($p < 0.05$, Wald-test, DESeq2) showing enrichment for synaptic processes (two-sided Fisher exact test). The test statistics are presented in Supplementary Data 10. **b** Heritability enrichment for schizophrenia after excluding the 193 genes with SynGO annotation. The $p$-values calculated from standard errors derived from block Jackknife as implemented in LD score regression. **c** *MEF2C* is upregulated in NPC of 22q11.2 deletion carriers (N(cases)= 19, N(controls)= 29, Wald-test from DEseq2). Data is presented in a Tukey-style boxplot with the median (Q2) and the first and the second quartiles (Q2, Q3) and error bars defined by the last data point within $+/-$ 1.5-times the interquartile range. **d** Spike count (mean number of spikes in a 10 s period). The activity of neurons derived from control (green, $N = 12$ lines, 104 wells) is compared to neurons from cases with 22q11.2 deletion ($N = 9$ lines, 54 wells). Pooled data is presented as Tukey-style box plots (Q1, Q2, & Q3; whiskers extend to the most extreme data points between Q1-1.5*IQR and Q3+1.5*IQR). Significance is determined by Kruskal-Wallis test with a 5 % significance level (* = $p$ value ≤ 0.05). **e** Proportion of electrodes detecting spontaneous activity, against the number of days post-induction. **f** The most weighted subcluster graph for protein-protein interactions (PPI) for differentially expressed genes in neurons.

---

*MEF2C* transcription factor, a regulator of the *JUN/FOS* signaling pathway, which in turn might reduce synaptic transmission.

Taken together, these results indicate that activity-dependent gene expression is changed in 22q11.2del cells, likely impacting downstream protein homeostasis and synaptic transmission. The results of our gene ontology and PPI analyses thus converge on the same key pathways that are regulated by 22q11.2del in each cell type. These results further demonstrate that the cell type-specific effects of the deletion involve distinct biological functions that may have clinical relevance for the phenotypic presentation in patients.

**Reduced network activity in 22q11.2del neurons**. Overall, our data suggests that changes linked to 22q11.2del during the development of excitatory neurons alter the balance of the *JUN/FOS* transcriptional pathway, which has well-established roles in activity-dependent gene expression[47]. We thus hypothesized that the transcriptional activation of this pathway and its targets, which plays a role in reducing synaptic transmission upon sustained activity[47] might result in decreased network activity in neuronal cultures with 22q11.2del.

We thus asked whether 22q11.2del neurons exhibited changes in network activity. Previously, we had shown that by 42 days of excitatory differentiation, neurons derived from control cell lines were spontaneously active and that their rate of firing was governed almost entirely by network activity mediated through synaptic connectivity[34]. We used multielectrode arrays (MEAs) to monitor neuronal network development and activity over 42 days of neuronal differentiation[34]. In 22q11.2del neurons, we detected a significantly lower spiking rate from 21 days of differentiation and onward, when compared to controls ($N$ = a total of 162 wells from 21 cell lines) (Fig. 4d, e). We found this result striking, as it was consistent with the notion that the altered abundance of synaptic transcripts and activity-dependent gene expression we observed by RNA sequencing was associated with functional effects on network activity in 22q11.2del neurons.

**Gene editing of the 22q11.2 deletion**. To complement our patient-driven study and assess whether 22q11.2del was sufficient to explain the transcriptional changes we observed in our patient-based discovery cohort, we used CRISPR/Cas9 to engineer the 22q11.2 deletion in a human embryonic stem cell line (H1/ WA01). Using guide RNAs that cut within the low copy repeats (LCRs) flanking the 3 Mb deletion, we generated heterozygous 22q11.2 deletion cell lines at a modest frequency (2/1000), as well as many non-targeted but otherwise isogenic controls (Fig. 5a-d). We then subjected the two deletion clones and two non-targeted control clones to neuronal differentiation and performed RNA sequencing on d0 hPSCs, d4 NPCs, and d28 excitatory neurons. In PCA, components one and two separated each of the samples by differentiation state, with the stem cell, NPC and neuronal cells showing strong reproducibility of differentiation across replicates (Fig. 5e). Components three and four then separated

the samples based on their deletion status, with 22q11.2del samples substantially separated from their non-targeted counterparts (Fig. 5f, Supplementary Fig. 9a). This separation was not solely due to deleted cis genes as it persisted upon removal of these genes from the PCA, indicating that it was a more global phenomenon in the transcriptome of the edited lines. Importantly, the genes driving the separation in PC3 and PC4 were largely shared by those differentially expressed in the discovery cohort. Out of the top 100 negative and positive loadings for PC3, 79 and 83, respectively, were nominally significantly changed also in neurons in the discovery cohort ($p < 0.05$). For PC4, this overlap was 39 and 60 out of 100, for negative and positive loadings, respectively.

We next performed differential expression analysis to delineate transcriptional changes present in clones edited to contain 22q11.2del (Supplementary Data 15–17). As expected, the edited lines showed systematic downregulation of genes in the deletion region at all stages ($p = 6 \times 10^{-61}$, Mann-Whitney test) (Fig. 5g) with 32, 26, and 29 deleted genes passing individually FDR < 5% cutoff in the isogenic hPSCs, NPCs, and neurons. Like the discovery set, *CAB39L* was consistently upregulated at all stages in lines with isogenic 22q11.2del. Overall, a highly significant number of genes exhibited aligned changes in transcript abundance between the discovery cohort and edited samples ($p < 0.05$ in both sample sets) across all differentiated stages analyzed: hPSCs, 69% (188 out of 273 $p = 4.2 \times 10^{-10}$, binomial test); NPCs, 82% (115 out of 141 $p = 1.5 \times 10^{-14}$, binomial test) and neurons, 76% (542 out of 712 $p = 5.6 \times 10^{-9}$, binomial test) with strongly correlated effect sizes ($r_{hPSC} = 0.58$, $r_{NPC} = 0.81$, $r_{neuron} = 0.58$, Pearson correlation); (Fig. 5h, Supplementary Fig. 9b, c).

We next wondered whether the pathways and cellular programs that were altered in a cell-type-specific manner in our discovery dataset were also altered in the edited lines. To this end, we examined the expression of genes contributing to the minimal PPI networks identified in the discovery dataset (Fig. 4g and Supplementary Fig. 8) and found that an overwhelming majority of these genes are changed in the same direction in cells with isogenic 22q11.2del at each stage, with 88%, 89 and 90% of the genes contributing to the PPI network in stem cells, NPCs and neurons respectively, being altered in the same direction in the isogenic and discovery datasets. Notably, the activity-dependent gene *MEF2C* was also increased in NPCs of H1 22q11.2del cells compared to isogenic controls (Fig. 5i, Supplementary Fig. 9d, e).

Furthermore, upon synaptic process annotation in SynGO we observed a replication of the induction of genes ($p < 0.05$) involved in synaptic vesicle cycle and endocytosis in the edited 22q11.2del neurons (GO: 0099504, $p_{FDR adj} = 0.0029$) (Fig. 5j, Supplementary Data 18). Overall, of the 193 transcripts with synaptic functions in the discovery dataset (Fig. 4a), 49 were also more abundant in neurons ($p < 0.05$) harboring the engineered 22q11.2del (Expected = 28 genes, $p = 8.2 \times 10^{-5}$, binomial test), out of which 21 passed the FDR < 5% cutoff for significance.

Additionally, the 82 transcripts implicated in the ubiquitination pathway that we found to be more abundant in 22q11.2del neurons

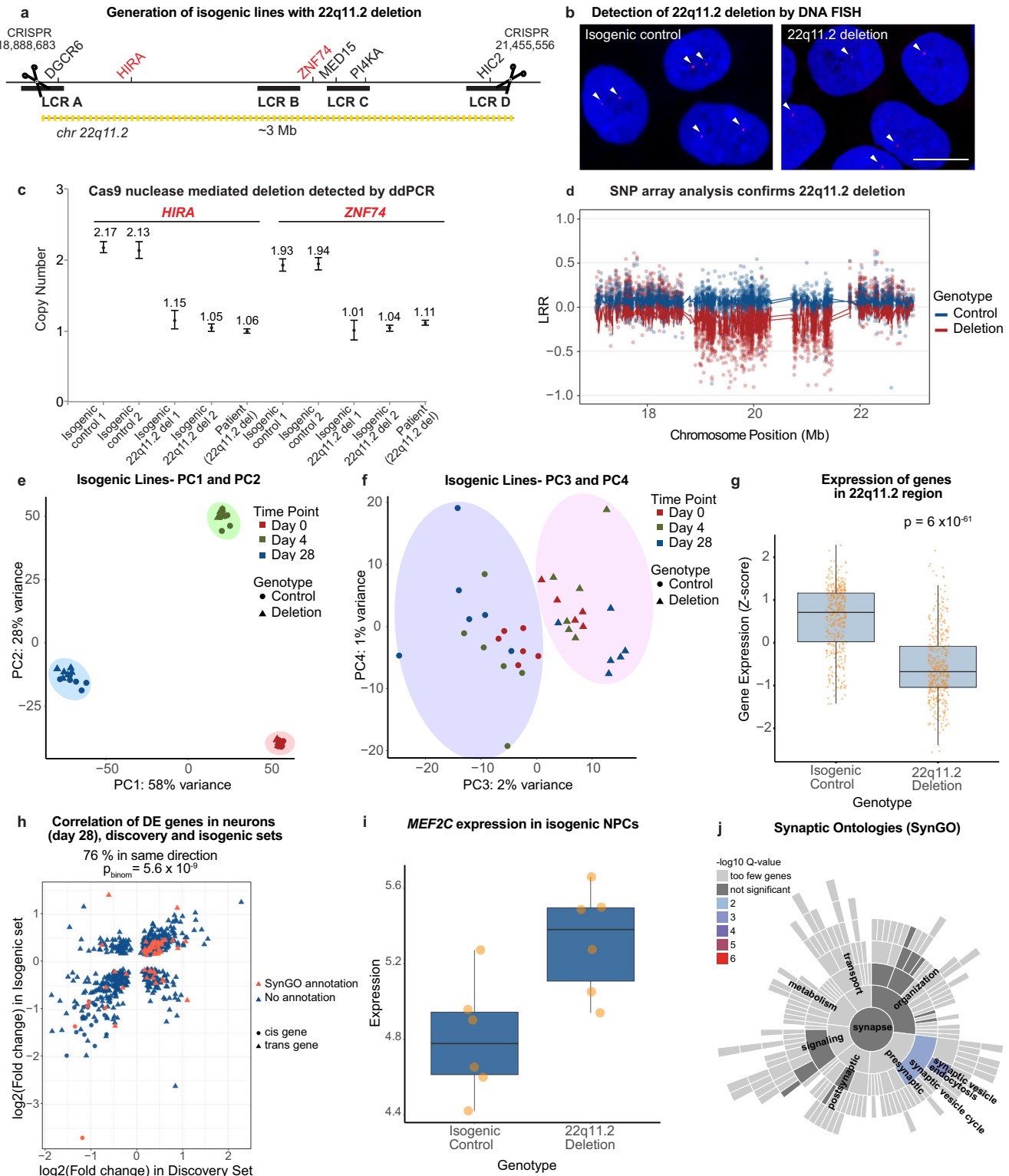

**a** Generation of isogenic lines with 22q11.2 deletion

**b** Detection of 22q11.2 deletion by DNA FISH

**c** Cas9 nuclease mediated deletion detected by ddPCR

**d** SNP array analysis confirms 22q11.2 deletion

**e** Isogenic Lines- PC1 and PC2

**f** Isogenic Lines- PC3 and PC4

**g** Expression of genes in 22q11.2 region

**h** Correlation of DE genes in neurons (day 28), discovery and isogenic sets

**i** *MEF2C* expression in isogenic NPCs

**j** Synaptic Ontologies (SynGO)

were on average 0.29 standard deviations (SDs) higher expressed in the edited lines (95%-CI:0.17–0.41 SDs, $p = 1.8 \times 10^{-6}$, $t$-test), with18 of these being individually significantly ($p$ value < 0.05) after gene editing ($p = 0.03$ binomial test) (Supplementary Data 19). Furthermore, 28 out of the 99 JUN target genes induced in the complete discovery dataset were also induced in neurons with isogenic 22q11.2del ($p < 0.05$) ($p = 0.00046$, binomial test, expected overlap = 14 genes). Finally, we examined the differentially expressed

genes ($p < 0.05$) for association to schizophrenia. Variants surrounding the induced genes in the edited lines revealed significant gene-wise association to schizophrenia consistent with the observation in the discovery cohort ($\beta = 0.11$, SE = 0.029, $p = 6.6 \times 10^{-5}$, $N = 1611$ genes, MAGMA). Thus, we conclude that 22q11.2del is indeed sufficient to explain most transcriptional effects we found to be associated with the deletion in our case-control cohort, including those related to the genetic risk for schizophrenia.

**Fig. 5 Validation of causality between differentially expressed genes and 22q11.2del in an isogenic setting. a** Generation of isogenic lines with 22q11.2del using CRISPR/Cas9 guide RNAs that cut within the low copy repeats (LCRs) flanking the 3 Mb deletion. Coordinates for the guides genomic position on chromosome 22 are indicated (Hg19). **b** Detection of isogenic 22q11.2del using DNA FISH analysis and a probe generated probe using CTD-2300P14 (Thermo Fisher Scientific, 96012). Blue = DAPI (DNA), Red=22q11.2 region. Scale bar: 10um. (N(edited clones) = 2 and N(nonedited clones) = 2; 3 experimental replicates each). **c** ddPCR assay to determine the copy numbers of *HIRA* and *ZNF74*, located in 22q11.2 (N=2 wildtype and 2 edited clones and 1 patient control). Analysis performed via QuantaSoft software (BioRad); copy number for *HIRA* and *ZNF74* (normalized to *RRP30*), error bars represent the Poisson 95% confidence limits. **d** SNP array marker intensity (LRR) for SNPs overlapping the deletion locus confirms 22q11.2del in two clones (red). **e,f** Principal component analysis of cell lines with and without isogenic 22q11.2del. Circles = genes within 22q11.2 (cis). Triangles = genes outside 22q11.2 (trans). **e** PC1 and PC2 separate cells by developmental stage., PC3 and PC4 separate cells by deletion genotype. **g** Significant downregulation of genes in 22q11.2 in lines with isogenic 22q11.2del (Mann-Whitney U test for 32 genes in 2 deletion and 2 control clones, two-sided). Data is presented in a Tukey-style boxplot with the median (Q2) and the first and the second quartiles (Q2, Q3) and error bars defined by the last data point within $+/-$ 1.5-times the interquartile range. **h** Correlation of fold changes in differentially expressed genes in discovery and isogenic datasets in neurons. Transcripts from 32 genes were detected and significantly changed in the discovery and isogenic lines (adjusted *p*-value < 0.05), of which nine were located outside 22q11.2 (*FAM13B, KMT2C, HYAL2, DNPH1, ZMYM2, VAPB, SMG1, CPSF4, MAP3K2*). All 32 genes were changed in the same direction in both cohorts ($p = 5.6 \times 10^{-9}$, binomial test). Genes with a SynGO annotation shown in red, genes with no SynGO annotation shown in blue. Circles = cis genes. Triangles = trans genes. **i** *MEF2C* is upregulated in 22q11.2del NPCs compared to isogenic controls (N(edited clones) = 2 and N(non-edited clones) = 2; 3 experimental replicates each). Data is presented in a Tukey style boxplot with the median (Q2) and the first and the second quartiles (Q2, Q3) and error bars defined by the last data point within $+/-$ 1.5-times the interquartile range. **j** SynGo annotation of genes induced in isogenic 22q11.2del neurons showing enrichment for synaptic vesicle cycle and endocytosis.

**Reduced pre-synaptic protein abundance in 22q11.2del neurons**. As an independent means of examining whether 22q11.2del impinged on presynaptic components in excitatory neurons, we performed whole cell proteomics on day 28 neurons from two patients and two controls (Fig. 6a) As expected, peptides mapping to genes within 22q11.2 were reduced in neurons harboring the deletion relative to levels in controls (Fig. 6b; Supplementary Data 20).

Importantly, consistent with the altered expression of activity-dependent genes and the reduced synaptically driven network activity in 22q11.2del neurons, we found that proteins downregulated in 22q11.2del neurons were enriched for synaptic gene ontologies (Fig. 6c). In total, 182 of the proteins that were downregulated in deletion carrier neurons had SynGO annotations. Of these, 37 were upregulated at the transcriptional level. Additionally, 31 proteins were upregulated in deletion carrier neurons and had SynGO annotations; 4 of which were also upregulated at the mRNA level (Supplementary Fig. 9f).

The synaptic components exhibiting alterations in deletion neurons were predominantly presynaptic and specifically involved in synaptic vesicle cycle ($p_{\text{FDR adj}} = 3.5 \times 10^{-19}$) (Fig. 6c; Supplementary Data 21), and included Synaptotagmin 11 (SYT11), Neurexin-1 (NRXN-1), and Synaptic Vesicle Glycoprotein 2 A (SV2A). SV2A (Fig. 6d) regulates vesicle exocytosis into synapses and works in presynaptic nerve terminals together with Synaptophysin and Synaptobrevin[67,68]. This finding converges with genetic studies as rare variants in SV2A have been shown to be significantly associated with schizophrenia[63,69]. Similarly, *NRNX1* has established roles in schizophrenia[63,70,71] and *SYT11*, located on the chromosome locus 1q21-q22 may be a risk gene for schizophrenia[72]. We confirmed the decreased expression of SV2A (Fig. 6e), along with the reduction of protein levels of SYT11 (Supplementary Fig. 9g) and NRXN1 (Supplementary Fig. 9h, 10) in 22q11.2del neurons by immunostaining or immunoblotting. Additional proteins with schizophrenia rare variant associations (via the SCHEMA consortium[62–64]) altered in 22q11.2del neurons included DNM3, MAGI2 and TRIO (downregulated in patient neurons) and HIST1H1E, SRRM2 and ZMYM2 (upregulated in patient neurons) (Supplementary Data 21).

## Discussion

Here we have explored the transcriptional and functional consequences of 22q11.2del on human neuronal differentiation (Supplementary Data 22–24). Our findings lead to several new insights into the biology of 22q11.2 deletion syndrome and how it confers risk for varied psychiatric disorders as neural development and differentiation unfold. We found that the genes whose expression is perturbed in 22q11.2del carriers connect the effects of the deletion to genes and pathways implicated in NDD, ASD and schizophrenia through large-scale exome sequencing and GWAS studies[27,28,32,44–46,62,64,73]. Thus, rather than working through independent mechanisms, our studies suggest the deletion confers risk for these various conditions at least in part by converging on the same gene products and pathways that are more widely disturbed in other patients.

Early during neuronal differentiation, we found that a significant number of the genes differentially expressed in 22q11.2del carriers had been previously linked to damaging or LoF sequence variants more widely identified in NDD and ASD. Interestingly, as differentiation proceeded and cells took on a post-mitotic, excitatory neuronal identity, the effects of 22q11.2del on the expression of genes outside the deletion lost enrichment for genes implicated in NDD/ASD and acquired an enrichment for genes harboring rare inactivating exome variants preferentially associated with schizophrenia, as well as genes in linkage disequilibrium with common genetic variants associated with schizophrenia, a result replicated using genotypic data from two independent GWAS studies. Just as signal from ASD/NDD-associated genes was absent in the neuronal stage of differentiation, the enrichment for effects on schizophrenia-associated genes was absent in stem cells and NPCs. This surprisingly selective signal is likely to reflect stage-specific cellular programs, such as synaptic processes being specific to neurons.

We found these transcriptional results striking as NDD and ASD are linked to biological processes acting early in brain development[74], while sequence variants associated with schizophrenia have been previously shown to be enriched for genes expressed in excitatory neurons and more recently for genes functioning in excitatory synaptic transmission[75]. Thus, we hypothesize that by looking in a human cell type with disease-relevant biology, we were able to identify previously unappreciated effects of 22q11.2del.

Our findings support human genetic studies suggesting that neuropsychiatric CNVs such as 22q11.2del likely interact with risk variants in the genetic background[23–25]. Transgenic mice carrying syntenic deletions have suggested neurodevelopmental abnormalities linked to the deletion or to individual genes within the

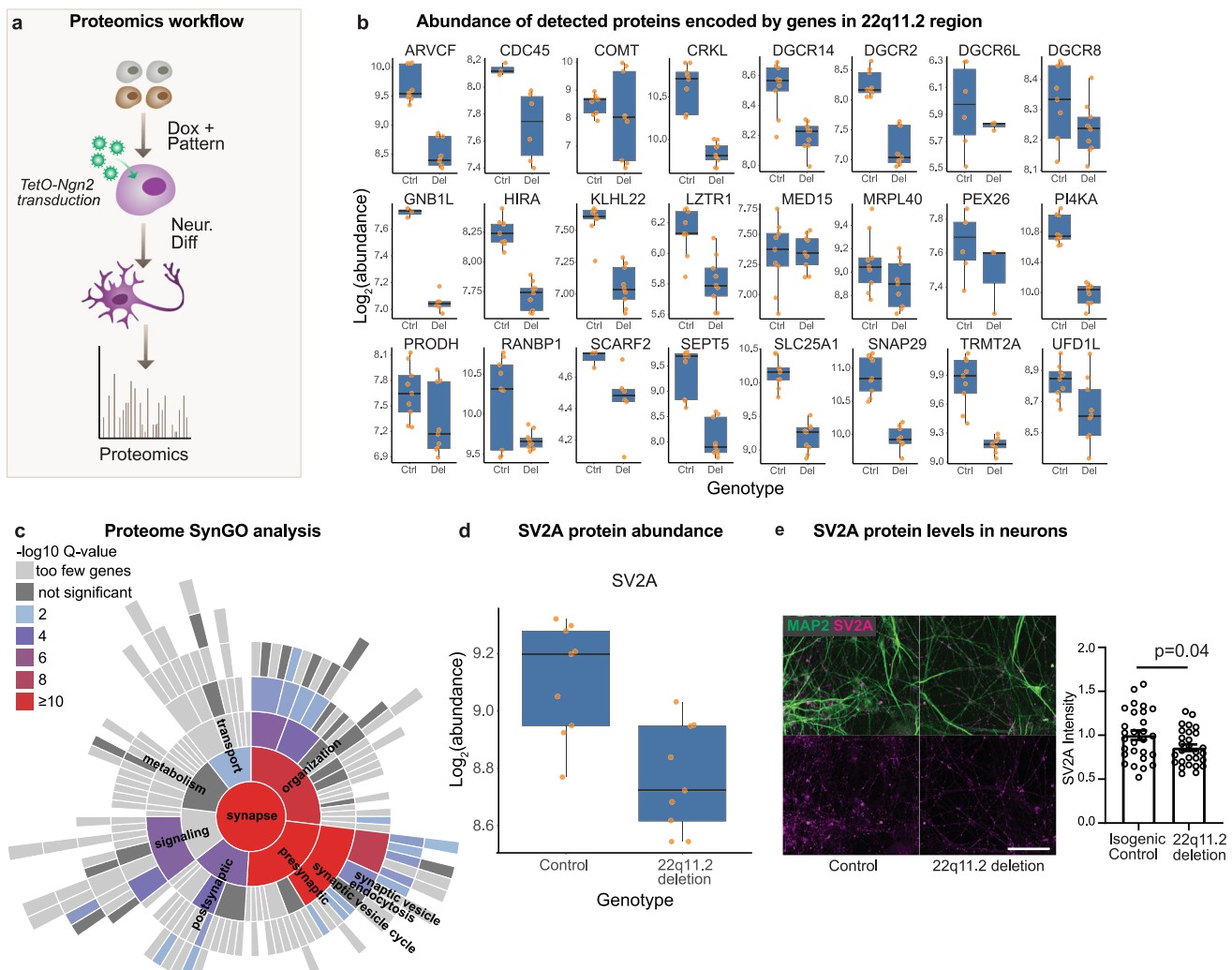

**Fig. 6 Whole cell proteomics on 22q11.2 deletion neurons. a** Workflow schematic. Neurons from deletion carriers and controls were harvested 28 days post neuronal induction. **b** Abundance of proteins encoded by genes in the 22q11.2 region detected by proteomics in neurons. Del = 22q11.2 deletion. Ctrl = control. (N(cases) = 2, N(controls) = 2, total 18 replicates). Data is presented in a Tukey style boxplot with the median (Q2) and the first and the second quartiles (Q2, Q3) and error bars defined by the last data point within +/− 1.5-times the interquartile range. **c** Synaptic gene ontologies (SynGO) in proteins downregulated in deletion carrier neurons. **d** SV2A protein levels detected by proteomics are decreased in deletion carrier neurons. N(cases) = 2, N(controls) = 2, total 18 replicates. Data is presented in a Tukey style boxplot with the median (Q2) and the first and the second quartiles (Q2, Q3) and error bars defined by the last data point within +/− 1.5-times the interquartile range. **e** SV2A protein levels detected by antibody staining are decreased in day 28 neurons derived from isogenic lines with 22q11.2 heterozygous deletion compared to controls. (Left) Representative confocal images of control and 22q11.2 deletion neurons stained with antibodies against SV2A (magenta) and MAP2 (green). Scale bar is 100 µm. (Right) Quantification of total SV2A fluorescence within MAP2-positive neurites normalized to isogenic controls. Data are means ± SEM. Individual points are analyzed fields of view from 4 independent inductions per condition. Statistical analysis by two-sided Student's *t*-test reveals statistically significant (*p* = 0.037) decrease in SV2A levels in deletion neurons.

region[13,20,76]. It is, however, important to keep in mind that such mice do not have genetic backgrounds harboring human polygenic risk alleles, which explain the majority of heritable variation in schizophrenia and other psychiatric phenotypes[46], and thus may fall short of reproducing human specific gene regulatory effects.

We used a new tool that we developed and report here to ask which minimal PPI networks best explain the changes in gene expression we observed. This analysis revealed that a surprising number of deletion components likely play a role in the transcriptional signals, in line with studies from mouse models[77]. In stem cells and NPCs the deletion impacts pathways linked to proliferation, NOGO signaling and RNA metabolism. In neurons, the deletion alters activity-dependent gene expression, protein homeostasis and presynaptic biology. *MEF2C*, an activity

dependent transcription factor and negative regulator of excitatory synaptic density[65,66] is overexpressed in 22q11.2del NPCs, likely due to the loss of one copy of *TBX1*, a known *MEF2C* inhibitor located in 22q11.2[48,49] (and which was not detected in our minimal PPI network). Increased expression of *MEF2C*, could lead to premature activation of the JUN/FOS pathway, which would be predicted to result in reduced network activity and synaptic connectivity.

To directly test this idea, we examined whether 22q11.2del neurons displayed reduced synaptic functionality. Using a network activity assay in these cells, which we have previously shown was largely driven by a mixture of AMPA and NMDA receptor-mediated transmission[34], we indeed found this to be the case. Many of the patients' neurons showed a significant

overall reduction in network activity relative to controls. Thus, the deletion was not only associated with induction of activity-dependent gene expression, but with aligned changes in neuronal function. Based on these findings, we would thus expect a decreased expression of synaptic proteins, which we do, indeed, detect.

Our proteomic examination of 22q11.2del neurons afforded an orthogonal examination of synaptic components in these cells and independently identified significant presynaptic alterations, including alterations in the schizophrenia-associated gene SV2A, a key mediator of pre-synaptic function. This is of translational and therapeutic importance given the existence of a positron emission tomography (PET) radiotracer specific for SV2A based on the drug Levetiracetam which now enables the in vivo investigation of presynaptic protein levels in the brain[78]. Interestingly, a recent PET-imaging study utilizing this SV2A radiotracer found a significant reduction in the abundance of SV2A in the cortex of schizophrenia patients relative to controls[79]. Genotyping of this schizophrenia patient population was not carried out prior to imaging and our results suggest that a more specific study examining SV2A levels in 22q11.2del carriers of varying diagnoses would be warranted.

Individual genes within 22q11.2 have been at the center of several studies aiming to identify causal genes underlying the syndrome. Many of these studies, using rodent, and more recently, human[40] models, have reported defects in synaptic processes and brain connectivity[80–82], many with a focus on *Dgcr8*, which encodes a subunit of the microprocessor complex which mediates microRNA biogenesis[13]. Khan et al[40] identified a calcium signaling defect in organoids containing mixed cell types derived from 22q11.2del and controls individuals, which could then be rescued by *DGCR8* overexpression. Whether these phenotypes can be recapitulated with a scaled sample set and defined cell types remains to be seen. Alterations in *DGCR8* might seem like a promising candidate for the distributed effects on gene expression we observed across many transcripts. However, reduced microRNA function from lower *DGCR8* copy number would predict an increased rather than decreased abundance of the synaptic proteins we found. Another candidate, *DGCR5*, which encodes a long non-coding RNA within 22q11.2, has previously been shown to regulate several transcripts encoding genes associated with schizophrenia[83]. However, that study found that reducing the function of *DGCR5* led to a reduction in the expression of its targets, again the inverse of our finding. Regulatory mechanisms can however be complex[84,85], and a causal role of these genes in the effects we observed cannot be completely ruled out.

A challenge in studying psychiatric conditions has been the difficulty to establish causal relationships between genetic variants of interest and their effects. In this study we utilized CRISPR/Cas9 to generate the 22q11.2 deletion in a control human stem cell line by inducing double strand breaks within the same repetitive elements (LCR22A and LCR22D) that are normally important mediators of the deletion. While the process was relatively inefficient (compared to the generation of isogenic lines with the shorter 16p11.2 or 15q13.3 deletions[86], for example), we were able to obtain two independent clones that carried this heterozygous deletion. Using these edited cells, we could then ask, without confounding by inherited variation elsewhere in the genome, which associations we had previously observed was the deletion sufficient to cause. We found that the deletion in this isogenic setting was sufficient to induce significant and aligned alterations in the expression of genes contributing to the minimal PPI network at each differentiation stage, including changes, in neurons, in activity-dependent, presynaptic, and proteasome pathways as well as heritability enrichment for schizophrenia.

When combined with genetic findings from 22q11.2del patients[23–25], our observations lead us to a model in which 22q11.2del exerts a strong effect on genetic risk factors for NDD and ASD genes early in differentiation, while in neurons the gene regulatory influence of the deletion shifts to risk factors for schizophrenia. Our gene editing experiments suggest that these distinct "pushes" on NDD/ASD and schizophrenia risk occur regardless of one's genotype.

How exactly 22q11.2del might regulate the expression of genes in trans remains a matter of great interest. One intriguing possibility is that 22q11.2del might impact chromatin architecture, thereby regulating the expression of genes outside of the region. Indeed, a recent study using 22q11.2del lymphoblastoid cell lines revealed changes in their genome architecture[87]. It is thus possible that 22q11.2del spatially rearranges the genome of neuronal cells, resulting in mis-regulation of genes linked to neuropsychiatric disorders.

The current study is not without its limitations. Even though it is, to our knowledge, one of the largest of the effect of 22q11.2del on human neuronal cells, our current sample size still falls short of enabling us to stratify the cohort by ancestry, diagnosis, age, sex, or deletion size. Additionally, most patients in our cohort have a schizophrenia diagnosis, along with an excess of male cases. Future studies with even larger sample sets could be sufficiently powered to enable the comparison of cells from 22q11.2del patients with different diagnoses, deletion sizes (such as the most common 3 Mb versus smaller nested deletions), age and sex.

Collectively, the novel iPSC lines, CRISPR edited cell lines, RNA sequencing data and functional phenotypes we report here will provide a framework for evaluating future therapeutic targets and candidates for 22q11.2del carriers. These carriers represent an interesting population for drug discovery as they are a group of individuals with more homogenous, yet still textured risk of psychiatric illnesses. We suggest that as aspects of the gene expression signal we observed are rescued, the functional relevance of such findings could be tested in the context of whether neuronal network activity is also restored in patient neurons. Through this approach, the likely multifaceted contributors to psychiatric illness that 22q11.2del confers could be quantitatively deciphered and the best approaches for alleviating its effects identified.

## Methods

Written institutional review board (IRB) approvals and study consent forms from each of the organizations contributing samples were sent to the Broad Institute of Harvard and MIT before the samples were sequenced and analyzed. All relevant ethical guidelines have been followed, and any necessary IRB and/or ethics committee approvals have been obtained. All ethical approvals are on file at the IRB office at Massachusetts General Brigham (MGB), formerly Partners, amended to protocol no. 2016P000058 'Cellular programming for neurobiological disease research'. This approval undergoes annual continuing review by the MGB Human Research Committee IRB. Supplementary review was conducted by the Broad Institute Office of Research Subject Protection.

**Human pluripotent stem cell (hPSC) lines cohort and derivation**. We assembled a scaled discovery sample set through highly collaborative, multi-institutional efforts with the Stanley Center Stem Cell Resource (Broad Institute), the Swedish Schizophrenia Cohort (Karolinska Institute), the Northern Finnish Intellectual Disability Cohort (NFID), Umea University, Massachusetts General Hospital (MGH), McLean Hospital, and GTEx. Human induced pluripotent stem cell (hiPSC) lines were generated from either fibroblasts or lymphoblasts, and either reprogrammed in house (as previously described[34]), at the New York Stem Cell Foundation (NYSCF) or at the Harvard Stem Cell Institute (HSCI) iPS core as listed in Table 1. The human embryonic stem cell (hESC) line H1 was obtained from the Human Embryonic Stem Cell Facility of the Harvard Stem Cell Institute.

**hPSC culture**. Human ESCs and iPSCs were maintained on plates coated with geltrex (life technologies, A1413301) in StemFlex media (Gibco, A3349401) and

passaged with accutase (Gibco, A11105). All cell cultures were maintained at 37 °C, 5% CO2.

**Infection of hPSCs with lentiviruses.** Lentivirus particles were produced by Alstem (http://www.alstembio.com/). hPSCs were seeded in a geltrex coated 12 well plate at a density of 100,000 cells/cm[2] in StemFlex medium supplemented with rock inhibitor (Y27632, Stemgent 04-0012) and lentiviruses, at a MOI (multiplicity of infection) of 2. 24 hours later, the medium was changed to StemFlex. The cells were grown until confluency, and then either maintained as stem cells, passaged, banked, or induced with Doxycycline for neuronal differentiation.

**Neuronal differentiation.** hPSCs were differentiated into cortical glutamatergic neurons as previously described[34]. Our protocol differs from previous Ngn2-driven protocols[33,88] through inclusion of developmental patterning alongside Ngn2 programming[34] (Fig. 1b, c, g). This paradigm generates post-mitotic excitatory cortical neurons that are highly homogeneous in terms of cell type[34] compared to most differentiation paradigms which yield heterogeneous cell types[89]. At 4 days post induction, cells are co-cultured with mouse glia to promote neuronal maturation and synaptic connectivity[90,91].

**RNA sequencing and alignment.** We used triplicate wells of each line at each time point to reduce experimental variation. Cells were harvested in RTLplus Lysis buffer (Qiagen 1053393) and stored at −80 °C. To minimize technical biases in readouts from cases and controls, we carried out the RNA sequencing in mixed pools of both genotypes. Sequencing libraries were generated from 100 ng of total RNA using the TruSeq RNA Sample Preparation kit (Illumina RS-122-2303) and quantified using the Qubit fluorometer (Life Technologies) following the manufacturer's instructions. Libraries were then pooled and sequenced by high output run on a HiSeq 2500 (Illumina). The total population RNA-seq fastq data was aligned against ENSEMBL human reference genome (build GRCh37.p13/hg19) using STAR (v.2.5)[92]. Prior to genome aligning, we used Trimmomatic (v.0.36)[93] to clip Illumina adapters and low-quality base-pairs from the ends of the sequence reads and removed reads with length < 36 base-pairs. The gene-wise read-counts were generated from the aligned reads by featureCounts in Rsubread (v.1.32)[94] using GENCODE GTF annotation version 19 (Supplementary Data 24). The reads from the three experimental replicates were summed together. The final read counts did not differ between cases and controls (11.0 × 10[6] and 10.8 × 10[6] reads, respectively; p = 0.68, two-sided t-test). The deleted cis genes accounted for 0.53 to 0.71‰ and 0.97 to 1.29 ‰ of all read counts in carriers and controls, respectively.

The plot in Fig. 2a was generated as follow: we used normalized read counts from DeSeq2 for a set of 18 canonical marker genes for pluripotency (SOX2 POU5F1, NANOG, and MKI67), neuronal progenitor cells (NEUROD1, SOX2, EMX2, OTX2, HES1, MSI1, and MKI67), and neuronal marker genes (RBFOX3, SYN1, DCX, MAP2, TUBB3, NCAM1, and MAPT) to address the progress of neuronal differentiation in the data set. The normalized gene-wise read counts were scaled to a standard score ($z = \frac{x-\mu}{\sigma}$) so that the gene expression of the different genes was presented as a difference from the average in units of standard deviations. The mean z-score for each gene set was then calculated and plotted as a line plot across the three cell stages (stem cells, NPCs, and neurons) with 95%-confidence intervals using inbuilt statistics in ggplot2.

**Differential gene expression analysis.** For differential gene expression analysis, we applied Wald's test for read counts that were normalized for library size internally in DESeq2[95]. The differential expression analysis was conducted separately for each cell stage to avoid any biases in gene variance modeling resulting from gene expression differences in between hPSCs, NPCs, and neurons. The experimental batch was included to the design formula in DESeq2 to correct for the 6 experimental batches in which the data was generated in. We used SVA package (version 3.32)[96] in R to search for latent factors to remove any unwanted variation in the data. We first estimated the number of latent factors using the leek method in num.sv function that was then used for calculating surrogate variables with irw method and five iterations in sva function. The design model for sva included experimental batch and deletion genotype. One latent factor was identified for the neuron data and was included to the design formula in DESeq2 for differential expression. For stem cells and NPCs no latent factors were identified. The results for differential expression were obtained for FDR adjusted p-value of <0.05. A principal component analysis was performed for all genes with more than 10 reads after normalizing the read counts by variance stabilizing transformation in DESeq2. For differential expression analysis in the edited isogenic deletion cell lines we used Limma-voom package[97,98] that enabled to model the non-independent experimental replicates from each clone with the "duplicateCorrelation" function, which was included in the design model by the block design in Limma.

**Power analysis.** The power estimates were calculated using RNASeqPower[99] (R package version 1.18.0). We calculated the median expression and variance in carriers and controls for all genes with one or more reads (25,264 genes) in the pilot data sets. We assumed equal number of cases and controls, while the coefficient of variance was calculated separately for cases and controls. The alpha level

was set to nominal significance of 0.05. For the final data set the power to detect fold changes of >2 was calculated for each gene separately.

**Enrichment for neurodevelopmental and constraint genes.** Gene lists for neurodevelopmental disorder genes were compiled from the deciphering developmental delay project[44,45], and recent large scale exome sequencing study in autism[28]. We included genes for which there was statistical overrepresentation of loss of function variants in patients compared to controls (total 97 genes for ASD[28] and 93 for ID[45] genes). From the earlier DDD-study[44] we included all "confirmed" developmental disorder genes that affect the brain. We included only those that had "hemizygous" and "monoallelic" as the allelic requirement, and mutation consequence defined as: "loss of function", "cis-regulatory" or "promotor mutation", and "increased gene dosage" (total 158 genes). This resulted in a list of total 295 disease genes for neurodevelopmental disorders (Supplementary Data 5). P-values for the enrichment analyses were calculated with hypergeometric test and binomial test in R. GO-term overrepresentations were calculated with hypergeometric test implemented in GoStats v. 1.7.4[100] in R with gene identifiers from org.Hs.eg.db. All p-values were calculated for overrepresentation using all mapped genes from each experiment as the background gene universe. False discovery rate (fdr) was used to adjust the raw p-values from the hypergeometric test for overrepresentation using p.adjust function in R. Significance threshold for overrepresentation was set to fdr-adjusted p-value smaller or equal to 0.05. The overrepresentation of synaptic GO terms was estimated by Fisher exact test in the SYNGO online portal (www.syngoportal.org) using a custom background gene set from the RNASeq data set.

**Protein-protein interaction network analysis.** Previous efforts have shown that the observed distribution of the p-values from differential expression studies could be modeled as a mixture of the distributed signal and uniformly distributed noise components[101,102]. In such approach, a threshold value could be estimated for observed p-values to discriminate between the likely true signal from noise. Hence, genes could be scored with logarithm of signal to noise ratio (log for making scores additive). Further, using a reference functional network we can leverage gene weights on the map of functional interactions to construct a node-weighted graph. Within this graph a search for the most-weighted connected subgraph (MWCS) could be performed. This search returns a functional module that has the strongest cumulative association to a trait being investigated. Appearance of genes in MWCS is driven both by their differential expression p-value and reference network topology. Thus, non-randomness of each gene's appearance could be evaluated by randomly permuting p-values and creating a random reference network with preserved node degrees. Estimates of how often a gene will be observed in MWCS by chance provide an empiric metric of significance and could be used to prioritize genes within MWCS. We implemented this strategy in R-package "PPItools" which provides a set of functions to identify MWCS, describe its statistical properties and prioritize genes within it. We used the InWebIM[52] direct protein-protein interactions network as a reference.

For every time point a beta-uniform mixture distribution was fitted to a distribution of observed p-values. Bonferroni adjusted significance threshold (0.05 / #Genes expressed) was selected as a threshold to discriminate positively and negatively scoring genes. Scores were estimated as a ratio between values of probability density function of Beta distribution at given p-value and threshold p-value or (α−1) × (log(x)-log(x_threshold)), where α is an estimated parameter of Beta distribution. MWCSs for every time point of the experiment (iPSC, neuronal progenitors and neuronal cells) were identified (Fig. 4g and Supplementary Fig. 8). Using described above permutational scheme, for every module we assessed a nonrandomness of presence for every gene found in the module (Supplementary Data 9). After multiple hypothesis testing correction (Bonferroni method used) several genes from each data set come up as functionally enriched (adjusted p < 0.05). 36 out of 50 genes in the iPSC module were seen in random MWCS with less than 5/1000 frequency.

We further tested for excessive connectivity between significantly differentially expressed genes and known neurodevelopmental disease genes. We selected 295 likely disease-causing genes from the Deciphering developmental delay (DDD) project, and a recent, large exome-sequencing study in autism (Supplementary Data 5). Curated inflammatory bowel disease (IBD) and Parkinson's disease (PD) risk gene lists (Supplementary Data 5) were included as a negative control set in this analysis. We estimated the number of connections between genes found in each of the disease gene lists and a list of differentially expressed genes with FDR <5% normalized to the total number of connections observed for all genes in both tested sets (disease and expression) in reference data. The obtained result could be interpreted as a proportion of all connections that are linking disease and differentially expressed genes. To evaluate significance, we generated random gene sets of the same size as the disease gene sets and estimated an expected number of connections with each set of differentially expressed genes. It is important to note that genes co-expressed within the same tissue or cell type tend to have a greater number of connections between them than would be expected for a random pair of genes. Hence, in generating random gene sets we specifically selected genes at random to match the expression pattern of a disease gene set in a given cell type (iPSC, neuronal progenitors or neurons). For every dataset, the expression distribution was binned into deciles and every gene was assigned to an appropriate

bin using mean counts. Random gene sets were selected to match the distribution of genes into deciles for disease gene sets. Empirical p-values were adjusted for two disease gene sets tested with Bonferroni correction.

The PPItools package for finding MWCS and performing network prioritizations along with documentation and source code to perform described analysis is available through GitHub[51].

**SNP heritability analysis.** LD Score regression[103] and MAGMA[59] were used for evaluating common variant associations in and near differentially expressed genes. Briefly for LD score regression, it can be shown that under a basic polygenic model we expect the GWAS statistics for SNP $j$ to be equation 1:

$$E[\chi_j] = N \sum_c \tau_c l(j,c) + 1$$

where $N$ is the sample size, $c$ is the index for the annotation category, lj,c is the LD score of SNP $j$ with respect to category $C_c$, and c is the average per-SNP contribution to heritability of category $C_c$. That is, the 2 statistic of SNP $j$ is expected to be a function of the total sample $N$, how much the SNP tags each category $C_c$ (quantified by lj,c, the sum of the squared correlation coefficient of SNP $j$ with each other SNP in a 1 cM window that is annotated as part of category $C_c$) and c, the effect size of the tagged SNPs.

With this model, LD Score regression allows estimation of each c. Each c is the contribution of category Cc after controlling for all other categories in the model (we included 74 annotations that capture different genomic properties including conservation, epigenetic markers, coding regions and LD structure similar to[104] and can be interpreted similarly to a coefficient from a linear regression. Testing for significance of c is useful because it indicates whether the per-SNP contribution to heritability of category C is significant after accounting for all the other annotations in the model. In addition to considering the conditional contribution of category $C_c$ with c, the total marginal heritability explained by SNPs in category $C_c$, denoted hg2(Cc), is given by equation 2:

$$\hat{h}^2(C_c) = \sum_{C:j\in C_c} , \sum_{\acute{c}:j\in C_{\acute{c}}} \hat{\tau}_{\acute{c}}$$

In other words, the heritability in category $C_c$ is the sum of the average per-SNP heritability for all SNPs in $C_c$, including contributions to per-SNP heritability from other annotations c' that overlap with category $C_c$ (as indicated by terms of the inner sum where c'≠c). Importantly, $\hat{h}_g(C_c)$ does not depend on the categories chosen to be in the model and provides an easier interpretation. Therefore, this quantity is the main focus of the analysis.

Here we focus on $\hat{h}_g(C_c)$ where $C_c$ comprises HapMap SNPs 100 kb upstream and downstream of each gene differentially expressed gene. $\hat{h}_g(C_c)$ was calculated for three sets of differentially expressed genes using two p-value thresholds (FDR < 5% and $p < 0.05$). Genes surpassing $p < 0.05$ cut-off were further divided to up and down-regulated genes. Heritability estimates were calculated for 6 sets of summary statistics from large GWAS of educational attainment[53] and 5 psychiatric/neurodevelopmental disorders: ADHD[54], autism spectrum disorder[55], bipolar disorder[56], major depressive disorder[57] and schizophrenia[58] OR[32]. In addition, the $\hat{h}_g(C_c)$ was calculated for the up-regulated genes in neurons ($p$ value < 0.05) and summary statistics for 650 phenotypes from the UK-biobank that have a significant heritability, defined by having a heritability $p$-value < 0.05 after Bonferroni correction for multiple testing (https://www.nealelab.is/uk-biobank/).

Similar to what was done for LD-score regression we considered gene-lists of differentially expressed genes to ask whether the differentially expressed genes are more strongly associated with each of the six phenotypes. We then used competitive gene set enrichment analysis using gene-wise p-values[55] that were calculated for each trait in MAGMA v 1.06 with standard settings[59]. All the results are adjusted for a set of baseline set of covariates with the goal to minimize bias due to gene-specific characteristics: gene size, log(gene size), SNP density, log(SNP density), inverse of the minor allele count, log(inverse of minor allele count) and number of exons in the gene. Gene-wise p-values were calculated by gene analysis in MAGMA and were used to identify genes underlying the stronger association signal among the upregulated genes in neurons. LD-score regression and MAGMA competitive gene set enrichment analyses were repeated for schizophrenia with 100 random genes lists that were matched with expression (±10%) to that of genes that were upregulated in deletion carriers in neurons.

**Analysis of enrichment of differentially expressed genes in whole-exome sequencing data.** We investigated if up- and down-regulated genes in 22q11.2del carriers are significantly disrupted by ultra-rare coding variants (URVs) in the whole-exomes of schizophrenia cases and controls (previously described[62,64]). In the cohorts separately, we regressed case status on the number of damaging URVs in the gene set of interest while controlling for the total number of URVs, sex, and the first five principal components. We define damaging URVs as putatively protein-truncating variants (stop-gain, frameshift, and splice-donor and acceptor variants), and damaging missense variants as variants with a MPC score of >=2, as previously described[105]. We applied inverse-weighted meta-analysis to combine the test-statistics from both studies to get a single joint P-value. We tested for

enrichment in up- and down-regulated genes, and a collection of randomly sampled neuronally-expressed genes.

**Motif enrichment analysis.** The motif enrichment analysis was carried out by Homer software for genes whose transcripts were found upregulated ($\log_2$ Fold change > 0) at day 28 neurons and $p$-value below <0.05. We performed a de novo motif analysis for human motifs using findMotifs.pl with len = 10. We curated the obtained results by setting a stringent p-value threshold ($p < 10^{-10}$), visually inspecting that observed motifs do not match only from the edges, excluded repeat sequences, and required that the motif had a frequency of above 5%.

**CRISPR generation of isogenic 22q11.2 cell lines.** To generate an isogenic 22q11.2 line in H1 hESCs, oligonucleotides (IDT) targeting LCR A (ACACTGGGCACATTATAGGG) and LCR D (CATTCATCTGTCCACCCACG) were cloned into a pU6-sgRNA vector generate sgRNA plasmids pPN298 and pPN306, respectively, via procedures described previously[106]. For transfection, cells were pre-incubated with "1:1 medium" composed of a 1:1 mixture of mTeSR1 medium and "hPSC medium" [hPSC medium: KO DMEM (Gibco 10829-018) with 20% KOSR (Gibco 10828-028), 1% Glutamax (Gibco 35050-061), 1% NEAA (Corning 25-025-Cl), 0.1% 2-mercaptoethanol (Gibco 21985-023) and 20 ng/ml bFGF (EMD Millipore GF0003AF) supplemented with 10 µM ROCK inhibitor (Y−27632). 7 µg Cas9 nuclease plasmid (pX459, Addgene #62988) 1.4 µg pPN298 and 1.4 µg pPN306 were electroporated into 2.5 × 106 cells at 1050 V, 30 ms, 2 pulses (NEON, Life Technologies MPK10096), as described[107]. Individual hPSC colonies were selected with puromycin treatment and seeded into Geltrex-coated 96-well plates, expanded for 1-2 weeks and duplicated for cell freezing and gDNA extraction. Clones were frozen in 96-well plates using 50% 1:1 medium plus 10 µM Y−27632, 40% ¬FBS (VWR SH30070.03) and 10% DMSO (Sigma D2650). gDNA was extracted overnight at 55 °C in Tail Lysis Buffer (Viagen 102-T) with Proteinase K (Roche 03115828001) followed by a 1 hr 90 °C incubation. Droplet digital PCR (ddPCR) was performed to determine copy numbers of the HIRA and ZNF74 genes. ddPCR is a fluorescence-based PCR assay where a PCR mix is partitioned into thousands of uniform droplets which, following amplification by a thermal cycler, are singularized and quantified for fluorescence intensity. The fraction of PCR-positive droplets is then analyzed using Poisson statistics for concentration of the template DNA in the original sample. By including a reference sequence in a second fluorescent channel, it is also possible to determine any copy number variation that might be present in the sample as well. Here, we used two pre-designed TaqMan probes (Bio-Rad) for genes that have been previously reported for the detection and validation of 22q11 deletion syndrome: HIRA (dHsaCP2500407) and ZNF74 (dHsaCPE5025806)[108], and compared them to a reference sequence RPP30 (dHsaCPE5038241). Genomic DNA from H1/WA-01 stem cells targeted with guides designed to remove 3MB from the 22q11 locus were extracted in 96-well format using the Quick-DNA 96 Plus Kit (Zymo Research), eluted in 30uL, and had an average concentration of 40 ng/uL. Twenty microliters of ddPCR sample was prepared using 4uL gDNA, 10uL ddPCR Supermix for Probes (no dUTP) (Bio-Rad), 1uL HIRA or ZNF74, 1uL control probe RPP30, 5U HindIII, and water. Droplets were generated using a QX100 Droplet Generator according to manufacturer instructions and run on a C1000 Touch thermal-cycler (Bio-Rad) using the protocol: 95 °C for 10 minutes, followed by 39 cycles of 94 °C for 30 seconds and 60 °C for one minute, then 98 °C for 10 more minutes and allowed to cool to 4 °C for at least 30 minutes before being run on a QX100 Droplet Reader. Data were analyzed using the QuantaSoft software with values representing the copy number value of each individual sample with error bars representing the Poisson 95% confidence limits. Baseline fluorescence was set by the analyst according to a no-template control well and the same baseline was applied to all samples. SNP genotyping was performed using the Illumina Infinium PsychArray-24 Kit on the lines to confirm the microdeletion (Broad Institute, Cambridge, MA). Differential expression for the isogenic lines was performed by DESeq2. The results from isogenic lines were compared to the results obtained from the discovery sample. The overlap between the direction of fold-changes in isogenic samples were tested using binomial test for all genes that were differentially expressed in the discovery sample. The expected probability for overlap was calculated from all genes and was on average 0.5. The differences in gene expression were tested by Mann-Whitney test including all genes with nominally significant p-value in differential expression in the isogenic lines.

**DNA FISH analysis.** FISH (Fluorescent In-Situ Hybridization) analysis was conducted in the isogenic control and 22q11.2del lines to analyze the copy number of the 22q11.2 region and validate the isogenic lines. We generated the probe using a bacterial artificial chromosome (BAC) located in the 22q11.2 region, CTD-2300P14 (Thermo Fisher Scientific, Supplier Item: 96012), labeled with Cy3 dUTPs (GE healthcare: PA53022), by means of nick translation (Abbott: 32-801300), and visualized the labeled cells using confocal microscopy.

**Multielectrode arrays (MEA).** MEA experiments and analysis were performed exactly as previously described[34]. Briefly, neuronal progenitors (at day 4) were seeded on 8 × 8 MEA grids, each with 64 microelectrodes, in the absence or presence of mouse glia, and routinely sampled these for 42 days after Ngn2

induction and dual SMAD and WNT inhibition. Each MEA plate contained wells from both deletion carrier and control neurons to minimize technical biases. Extracellular spikes (action potentials) were acquired using Axion Biosystems multi-well MEA plate system (The Maestro, Axion Biosystems; 64 electrodes per culture well). During the recording period, the plate temperature was maintained at 37 ± 0.1 °C, environmental gas composition was not maintained outside of the incubator. Unless otherwise stated, descriptive statistics for MEA data is presented as Tukey style box plots, showing the 1st, 2nd, and 3rd quantile (Q1, Q2, & Q3 respectively; inter-quartile range, IQR = Q3- Q1). Box plot whiskers extend to the most extreme data points between Q1-1.5*IQR and Q3+1.5*IQR[109–111]. All data points outside the whiskers are plotted. Non-parametric 95 % confidence intervals for M are calculated using fractional order statistics[112].

**TMT-processing workflow**. Cell pellets were lysed and 50ug protein per TMT channel were subjected to disulfide bond reduction and alkylation. Methanol-chloroform precipitation was performed prior to protease digestion with LysC/trypsin. Obtained peptides were labeled with the respective TMT reagents and pooled. Enhanced proteome coverage was achieved by high-pH reversed phase fractionation to reduce sample complexity. Peptide fractions were analyzed on an Orbitrap Fusion mass spectrometer using SPS-MS[113]. Mass spectra were processed using a Sequest-based in-house software pipeline. Peptide and protein identifications were obtained following database searching against all entries from the human UniProt database. For TMT-based reporter ion quantitation, we extracted the summed signal-to-noise (S:N) ratio for each TMT channel. For protein-level comparisons, peptide-spectrum-matches (PSM) were identified, quantified, and collapsed to a 1% peptide false discovery rate (FDR) and then collapsed further to a final protein-level FDR of 1%. Moreover, protein assembly was guided by principles of parsimony to produce the smallest set of proteins necessary to account for all observed peptides. Proteins were quantified by summing reporter ion counts across all matching PSMs using in-house software. Protein quantification values were exported for further analysis.

**Analysis of protein abundances**. Differences in protein abundances between deletion carriers and controls were estimated in day 28 neurons derived from two patient (SCBB-1962 and SCBB-1825) and two control lines (SCBB-1828, SCBB-1827) in total 18 replicates. The abundances for the detected 8811 gene products were log$_2$+1 transformed and quantile normalized in Limma package[98] (v. 3.3.49) in R. A linear model including instrument run and deletion status was used to analyze differences in the normalized protein abundances between deletion carriers and controls in Limma. The correlation of the non-independent experimental replicates was estimated with "duplicateCorrelation" function (average estimated inte-replicate correlation was 0.83) and was taken into account in the design model using block design in Limma. Overlap of gene products between RNA sequence data and proteomics data (total 8585 gene products detected by both methods) was compared using $p$ value < 0.05 threshold. The overlap of direction of effect was estimated with binomial test with expected probability of 0.5. The density coloring was calculated from Kernel density estimation using densCols in R.

**Immunohistochemistry**. Cultured induced neurons were fixed in 4% paraformaldehyde + 20% sucrose in DPBS for 20 min at room temperature. Cells were incubated with blocking buffer containing 4% horse serum, 0.1 M Glycine, and 0.3% Triton-X in PBS for 1 hour at room temperature. Primary antibodies, diluted in 4% horse serum in PBS, were incubated overnight at 4oC. Secondary antibodies were diluted in 4% horse serum and applied for 1 hour at room temperature. Samples were washed 3x with PBS and imaged on spinning disc confocal microscope (Andor Dragonfly) with a 20x air objective. The following antibodies were used: rabbit anti-SV2A (1:1000, Abcam ab32942), chicken anti-MAP2 (1:10,000, Abcam ab5392), rabbit anti-Synaptotagmin-11 (1;1000, Synaptic Systems 270 003), mouse anti-hNA (1:1000, Millipore MAB1281), rabbit anti-Cux1 (1:500, Santa Cruz m-222), guinea pig anti-NeuN (1:1000, Synaptic Systems 266004), mouse anti-Sox2 (1:500, R&D systems, MAB2018), mouse anti-Sox1 (1:500, R&D systems, AF3369), mouse anti-Ki67 (1:500, BD Biosciences 550609), rabbit anti-FoxG1 (1:500, abcam ab18259). Alexafluor plus-555 and Alexafluor plus-488 conjugated secondary antibodies (1:5,000) were obtained from Invitrogen.

**Image acquisition and analysis**. Fluorescent images were acquired on spinning disc confocal microscope (Andor Dragonfly) at room temperature using 20x air interface objective using Fusion software. For quantification at least four 1024 × 1024 pixel fields of view from 2 different wells were taken for each line. The images were analyzed using ImageJ (v1.52p) software. For colocalization analysis, Dapi-positive nuclei were counted for NPCs and hNA-positive nuclei were counted for D28 neurons. The percentage of cells colocalized with each neuronal or NPC marker was calculated in ImageJ.

**Immunoblotting**. For collection, neurons grown on glia were washed with DPBS and lysed with RIPA buffer and 1x protease inhibitor cocktail. Lysates were boiled, sonicated and centrifuged at 16,000xg for 5 minutes. The soluble fraction was separated on SDS-PAGE using Bolt system (Novex). The proteins were transferred onto nitrocellulose membrane using iBlot2 Gel Transfer Device and immunostained using Neurexin-1 antibody (Millipore ABN161-I, 1:1,000), Tuj1 (Biolegend 801201, 1:5,000), MEF2C (Abcam ab211493, 1:200 dilution, EPR19089-202) and GAPDH (Proteintech, 60004-1-Ig, 1:5000). and detected via HRP-conjugated secondary antibodies on the Chemidoc system.

**qPCR analysis**. RNA isolation was performed with the Direct-Zol RNA miniprep kit (ZYMO: cat# R2051) according to the manufacturer's instructions. To prevent DNA contamination, RNA was treated with DNase I (ZYMO: cat# R2051). The yield of RNA was determined with a Denovix DS-11 Series Spectrophotometer (Denovix). 200 ng of RNA was reverse-transcribed with the iScript cDNA Synthesis Kit (Bio-Rad, cat# 1708890). For all analyses, RT–qPCR was carried out with iQ SYBR Green Supermix (Bio-Rad, cat# 1708880) and specific primers for each gene (listed below) with a CFX384 Touch Real-Time PCR Detection System (Bio-Rad). Target genes were normalized to the geometric mean of control genes, RPL10 and GAPDH, and relative expression compared to the mean Ct values for control and wild-type isogenic samples, respectively.

The following primers were used:
MEF2C_forward 5′-CTGGTGTAACACATCGACCTC-3′
MEF2C_reverse 5′-GATTGCCATACCCGTTCCCT-3′
TBX1_forward 5′-ACGACAACGGCCACATTATTC-3′
TBX1_reverse 5′-CCTCGGCATATTTCTCGCTATCT-3′
RPL10_forward 5′-GCCGTACCCAAAGTCTCGC-3′
RPL10_reverse 5′-CACAAAGCGGAAACTCATCCA-3′
GAPDH_forward 5′-GGAGCGAGATCCCTCCAAAAT-3′
GAPDH_reverse 5′-GGCTGTTGTCATACTTCTCATGG-3′

**Reporting summary**. Further information on research design is available in the Nature Research Reporting Summary linked to this article.

## Data availability

All summary-level data generated in this study are available in Supplementary Data 22 and 23. The raw sequence datasets generated for the study are not yet deposited in a public repository due to varied consent provenance within our selected cohort and subsequent data access restrictions. Subsets of the data will be made available by the corresponding authors upon reasonable request within a 30-day timeframe under a data transfer agreement. The following human reference genome was used: ENSEMBL/GRCH37/Hg19 (http://ftp.ensembl.org/pub/grch37/). A reporting summary for this Article is available as a Supplementary Information file.

## Code availability

Computer code relevant to the PPI analysis has been deposited in GitHub[51]. All other analyses were performed using publicly available software and specified in the methods section. Computer code and data analysis will be made available upon request.

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

## Acknowledgements

We thank the many donors, institutions, and investigators world-wide that provided their cell lines and supported the publication of the results. We are indebted to Maura Charlton, Genevieve Saphier and Kristen Elwell for their assistance with the regulatory and logistical efforts required to acquire and sequence hiPSC lines. We regret the omission of any relevant references or discussion due to space limitations. The Genomics Platform at the Broad Institute performed sample preparation, sequencing, and data storage. This work was funded predominantly by U01MH105669 (NIH/NIMH) to MD and KE, with additional support from the Stanley Center for Psychiatric Research at the Broad Institute, R37NS083524 and U01MH115727 to SM, KE and RN, a NARSAD young investigator award (Brain and Behavior Research Foundation) and a Bn10 grant (Broad Institute) to RN, Sigrid Juselius Foundation, Orion Research Foundation, Instrumentarium Science Foundation, and Päivikki and Sakari Sohlberg Foundation awards to OP, and support from the Ministry of Science and Higher Education of the Russian Federation (Agreement No. 075-15-2022-301) to AL.

## Author contributions

R.N., O.P. and K.E. conceived the work, designed the experiments, analyzed the data and wrote the manuscript. R.N. supervised and performed the experiments, with help from A.T., C.B., M.T., R.M., E.J.G., V.V., D.H., E.P., and E.Z, and equipment provided by SF. O.P. performed the computational analysis, with help from M.T. and G.G. M.A. performed the PPI analysis, with help from A.L. and supervision from M.D. C.B. performed the proteomics experiments with help from J.A.P. and supervision from J.W.H. L.L. and A.G. carried out the SNP heritability analysis, with oversight from B.N. T.S. carried out the rare variant analysis. J.S. performed the MEA analysis. D.M., A.B., A.M.B. and D.Z.H. carried out the CRISPR editing, supervised by L.E.B. A.N. and C.L. assisted with stem cell compliance and data deposition. O.K., E.H., and M.K. provided the NFID cell lines, with oversight from A.P. C.M.H. and A.K.K. contributed the KI cell lines. B.C. and D.M. provided cell lines from Mclean Hospital. J.M., R.A. provided the Umea samples, R.D. provided the Stanford cell lines, and R.P. provided the MGH cell line. S.M. and S.H. provided guidance throughout the project.

## Competing interests

K.E. is Group Vice President, Head of Research and Early Development at Biomarin Pharmaceuticals and a founder of Q-state Biosciences, Quralis and Enclear. J.W.H. is a founder and advisor of Caraway Therapeutics. The remaining authors declare no competing interests.
