## [Peer Review File · Nature Communications]

The 22q11.2 region regulates presynaptic gene-products
linked to schizophreniaEditorial Note: This manuscript has been previously reviewed at another journal that is not operating a transparent peer review scheme. This document only contains reviewer comments and rebuttal letters for versions considered at *Nature Communications*.

REVIEWER COMMENTS

Reviewer #1 (Remarks to the Author):

22q11.2 deletion confers the largest effect of known genetic risk factor for schizophrenia. The manuscript entitled “the 22q11.2 region regulates presynaptic gene-products linked to schizophrenia” by Nehme et al. reports that 22q11.2 deletion impacts genes and pathways that may converge with risk loci implicated by psychiatric genetic studies using neural cells derived from 22q11.2 deletion carriers. Although the topic is of great interest to the community, the manuscript in its current form is poorly organized and the mechanisms are not well-explored. The authors should cut down the texts and reorganize the manuscript to make the main points clear to the readers. Moreover, the manuscript would benefit from additional validation experiments. I've detailed my comments below.

1. The authors used a differentiation protocol that relied on Ngn2 overexpression. Will Ngn2 overexpression confound the detection of gene expression differences between the groups?
2. The differentiation process is not well characterized. It is unclear whether the iPSCs could differentiate to neural progenitor cells within 4 days. The authors need to perform immunohistochemistry for marker genes and characterize the process. What percentage of cells express NPC markers at 4 days and what percentage of cells express excitatory neuronal markers at 28 days? Do all the lines have similar percentage of NPCs and neurons?
3. To help the readers better understand the function of DE genes identified in Figure 2, the authors could perform GO analysis with the DE genes identified at different time points.
4. It is unclear why the authors compared their DE genes to genes in ASD database? There is only one patient with ASD diagnosis.
5. It is very confusing which DE gene list was used in Figure 4. There are a total 133 DE genes in neurons identified in Figure 2b but the authors reported using 239 of the 2864 transcripts with increased abundance for the analysis in Figure 4. The authors need to explain clearly how they curated the data for this analysis.
6. The authors developed a new tool (PPItools) for analyzing protein-protein interaction networks. The authors should compare this tool to other tools like STRING and explain how it can improve the analysis.
7. The authors stated that many genes in the 22q11.2 interval interact with JUN/FOS transcriptional pathways. Can the authors experimentally explore how it could contribute to disease risk? Can the authors rescue any phenotypes by manipulating this pathway in the isogenic deletion lines?

Reviewer #2 (Remarks to the Author):

The authors have prepared a very thoughtful and extensive revision. The authors did further experiments to validate RNAseq and proteomic hits. This revision is now suitable for publication and a significant addition to the literature.

Reviewer #3 (Remarks to the Author):

The authors have evidently done a great deal to address many of the issues raised in the original set of reviews, and the manuscript overall has been improved.

However, the revised manuscript continues to include one iPSC cell line that has no 22q11.2 deletion. This 134 kb common variant within the low copy repeat A (LCR22-A) sequence, Karolinska sample SCBB-1430 from a 66 year old patient with schizophrenia. The fact that there are interesting and important results despite the erroneous inclusion of this cell line does support the power of the remaining sample, but for the authors not to recognize that this cell line does not belong in a “22q11.2 deletion” group, and indeed now to repeatedly call this a “nested 22q11.2 deletion”, is unacceptable.

There is another 22q11.2 deletion included (SCBB-1961, Stanford) that historically would not have been detected using a typical FISH probe, but would at least be called pathogenic using clinical microarray technology, and could be considered a “nested 22q11.2 deletion”, although the breakpoints provided indicate that it would not be included in most studies of the 22q11.2 deletion syndrome as these usually rely on the commonly deleted region that includes HIRA (TUPLE1 probe), and involve the recurrent LCR22A-LCR22B and LCR22A-LCR22C deletions that are nested within the LCR22A-LCR22D deletion region. The 2.5-3.0 Mb LCR22A-LCR22D deletion is the most common of the collection of rare, recurrent, pathogenic 22q11.2 deletions with replicated high risk of schizophrenia, and appears to be the 22q11.2 deletion extent for the other 18 cell lines used in the sample here, including the two used in their “pilot” study.

This remains an important point for the authors to address, given that the “nested 22q11.2 deletion” that is represented by many engineered mouse models is closest to the recurrent LCR22A-LCR22B human 22q11.2 deletion (not present in any of the samples used here). The manuscript would benefit from demonstration that the authors understand these basic issues; most interested readers will expect the breakpoints of the 22q11.2 deletions to be made explicit using standard nomenclature (LCR22A-LCR22D) in order to appreciate the significance of the findings.

In fact, there is some emerging evidence that smaller nested LCR22A-LCR22B 22q11.2 deletions may have a somewhat less severe neuropsychiatric phenotype, at least with respect to intellect (see e.g., Zhao et al., 2018 PMID: 30289625). The older references presented to attempt to support inclusion of a unique nested variant 22q11.2 deletion (and the erroneous non-deletion) cell line are superseded by this reference which reports data for a large enough sample to be able to detect phenotypic differences.

The remainder of the manuscript implies that there should be “...‘the’ 50% decrease in expression of ‘the hemizygote’ genes found in the (22q11.2) deletion region” (p 8).

Given that 12 of the cell lines used were from individuals with schizophrenia, with three to four others too young to know outcome, this means that 12 to 16 of the 19 cell lines could be from individuals with 22q11.2 deletion and schizophrenia or related psychotic illness. Possible implications of this ascertainment bias on the findings should be presented in the Discussion.

It would be useful to point out that Parkinson’s disease, while likely to be “unrelated” to findings in excitatory neurons, is a known associated feature of 22q11.2 deletions, thus would not be considered “unrelated” to the condition under study.

Data and methodology

While the sections on the MAGMA and SCHEMA variant findings have been changed, I find that the authors’ explanation of the findings and differences between these in the response to reviewers letter

is clearer than that in the manuscript; the same for the SynGO findings.

I would expect that GO (gene ontology) and PPI analyses would have very similar origins, thus it would not be surprising for results to converge.

The Discussion, while improved and now containing mention of some limitations, continues to need improvements with respect to placing the study results in the context of the existing literature. For example, it is consistent with evidence from other studies that the 22q11.2 deletion confers risk for various phenotypes at least in part through pathways known from other individuals without this deletion. It is not surprising that genes (? "components") in the 22q11.2 deletion region play a role in transcriptional signals - the mouse literature is replete with evidence of this. The results of the study suggest that the 22q11.2 deletion may have an impact on pathways linked to proliferation, etc.

A note about TBX1 not appearing in any of the PPI results presented would be welcome, particularly given the speculation about its role with respect to MEF2C.

Clarification that the authors found some enrichment of expression of genes in the 22q11.2 deletion excitatory neurons studied that in other studies of schizophrenia without a 22q11.2 deletion show evidence of rare loss of function variants, would be welcome.

There is not a perfect correlation between mRNA and protein levels, as the study results support, with the likelihood that complex regulatory mechanisms are involved, including those that may be affected by decreased DGCR8 gene dosage, and not always in readily appreciated / simple directions. Would the authors consider this, and also that the strongest predictor of protein level for any given gene is the rate of translation, followed by mRNA levels (see PMIDs: 21593866, 30694346). It would seem premature to exclude any of the genes in the 22q11.2 deletion region from a role in the findings of this study (p 30, 32).

The authors need to specify which of "the same repetitive elements" (p 31) were involved in the engineered human 22q11.2 deletion iPSCs, and relate to "this large structural variant", and "which associations...the deletion (was) sufficient to cause" (presumably the low copy repeat sequences A and D flanking the typical 22q11.2 deletion region). This is also very important as it relates directly to the other iPSCs studied that were derived from the 18 individuals included in the study who had LCR22A to LCR22D 22q11.2 deletions. While the CRISPR/Cas9 generated lines very much implicate the importance of the effects of this 2.5 Mb 22q11.2 deletion on the rest of the genome including factors involved in NDD/ASD and schizophrenia risk, these effects would presumably remain affected by the background genome (p 31) of this male line with no phenotypic information.

2.5 Mb of 51 Mb would not generally be considered consistent with saying "The 22q11.2 deletion lacks a large portion of chromosome 22..." even if only the most common LCR22A to LCR22D 22q11.2 deletion was being considered here.

Specific discussion of the potential effects of the 22q11.2 deletion size (including as noted above), diagnosis of most originating individuals with schizophrenia, and male excess of cases, needs to be included in the limitations section. If there were data presented in the results related to age or sex, as hinted at in the Discussion (p33), I did not see these.

Clarity and context:

Wording in the abstract sentence beginning "In more differentiated excitatory neurons," about the results related to indirect association needs to be clearer lest readers interpret that the variants assessed were those in the iPSCs. Throughout the manuscript, care in use of the terms "mutant" or "mutations" is warranted when the authors really mean "variants". The same goes for "the deletion"

when they mean “effects of the 22q11.2 deletion”. And (p 19) “In comparison, genes...” with reduced expression by the presence of the 22q11.2 deletion in excitatory neurons... would appear more accurate than what is stated. More judicious use of definitive statements e.g., “are” when the authors mean “may be” (e.g., p 17), and of terms such as “strikingly” would be welcome (e.g., p 14). This is not a “patient-driven study”.

There continues to be some evidence of lack of proof-reading and attention to detail, e.g., “in road” (p 3), “seven genes” then list only 6 (p 8), “falls shorts” (p32). There also needs to be some attention to ensure that the text and figures match with respect to gene names used (e.g., TRKA receptor, p 17). Also, in figures there is still a need to change to white font if the font is on a dark background for readability.

NCOMMS-21-37746-T Response to referees

Re: "The 22q11.2 region regulates presynaptic gene-products linked to schizophrenia"

We thank the reviewers for their careful consideration of our manuscript. Below each comment, we detail, point by point, how we have addressed it, in blue. As part of the submission we have also provided, in addition to a revised "clean" version, a version where all changes we have made to the text are marked.

REVIEWER COMMENTS

Reviewer #1 (Remarks to the Author):

22q11.2 deletion confers the largest effect of known genetic risk factor for schizophrenia. The manuscript entitled "the 22q11.2 region regulates presynaptic gene-products linked to schizophrenia" by Nehme et al. reports that 22q11.2 deletion impacts genes and pathways that may converge with risk loci implicated by psychiatric genetic studies using neural cells derived from 22q11.2 deletion carriers. Although the topic is of great interest to the community, the manuscript in its current form is poorly organized and the mechanisms are not well-explored. The authors should cut down the texts and reorganize the manuscript to make the main points clear to the readers. Moreover, the manuscript would benefit from additional validation experiments. I've detailed my comments below.

We are glad that the reviewer finds our work of great interest, and we appreciate the thoughtful comments and suggestions, which we have addressed below. Additionally, during this revision and the previous one, we have reorganized the manuscript to clarify and streamline the main message, including shortening / clarifying the text, removing some redundant Figure panels, and including validation experiments, as the reviewer recommended.

1. The authors used a differentiation protocol that relied on Ngn2 overexpression. Will Ngn2 overexpression confound the detection of gene expression differences between the groups?

We agree with the reviewer that this is an important point to clarify. To address this question, we have examined the expression of Ngn2 in all samples in our cohort, both at day 4 (at the NPC, Neuronal Progenitor Cell stage) and at day 28 (at the neuronal stage) (Figure 1 below). We found that overall, there were no significant differences in Ngn2 expression between cases and controls at either day 4 (difference = 0.2 log₂(CPM), 95%-CI:-0.35 – 1.44, p=0.22) or day 28 (difference= -0.13 log₂(CPM), 95%-CI:-1.70 –1.45 , p=0.8).

Figure 1. Ngn2 expression levels are not significantly changed across control and 22q11.2 deletion groups.

2. The differentiation process is not well characterized. It is unclear whether the iPSCs could differentiate to neural progenitor cells within 4 days. The authors need to perform immunohistochemistry for marker genes and characterize the process. What percentage of cells express NPC markers at 4 days and what percentage of cells express excitatory neuronal markers at 28 days? Do all the lines have similar percentage of NPCs and neurons?

We thank the reviewer for giving us another opportunity to further clarify this point. As we now emphasize in text, we have previously thoroughly characterized the neurons generated via the neuronal differentiation approach utilized here and reported our findings in a previous publication (Nehme et al 2018¹), and with subsequent publications in Wells et al 2018², Fan et al 2018³, Mitchell et al 2020⁴, and Wells et al 2021⁵. In our previous response letter, we have detailed our main findings from these characterization efforts (including immunostaining data, qPCR, RNA sequencing and comparison to published datasets such as Brainspan). In brief, we have examined and published data from multiple cell lines (>100) using this protocol prior to the current work, and the identity of the cells is the same across all cell lines: NPCs at day 4, and excitatory neurons (expressing upper layer cortical markers) at day 28. In the current study, we do not detect any differences with respect to the subtype of neurons generated, and cells from either 22q11.2 deletion donors or controls do not cluster separately in PCA at any of the examined cellular changes (stem cells, NPCs or neurons) (Fig. 1i of the manuscript), providing further evidence to this point.

Here, to further support this point, and on the reviewer's recommendation, we have performed immunohistochemistry for marker genes of neuronal progenitor cells (FOXP1, NESTIN, SOX1 and SOX2) at day 4, and marker genes of [excitatory] neurons (CUX1, NEUN, and MAP2) at day 28, on cells from both control and 22q11.2 deletion lines and quantified the percentage of cells expressing these markers. We have now added this data in a **new Figure panel, Extended Data Figure 2a**. Taken together, these data show that regardless of genotype, cells

indeed express progenitor markers at day 4, and neuronal markers at day 28, and all lines have a similar percentage of NPCs and neurons at these stages.

Should it be helpful to the reviewer, we reproduce below some of the key experiments and analysis from previous publications characterizing the cells generated by this approach (reproduced from our previous response letter). In summary:

- This differentiation approach gives rise to a homogenous population of excitatory neurons across all lines examined (Nehme et al 2018, Wells et al 2018, Mitchell et al 2020).
- The excitatory neurons that are generated resemble upper layer cortical excitatory neurons, expressing CUX1, CUX2, BRN2, and SATB2.
- These neurons also express AMPA and NMDA receptors, which are functional as evidenced by pharmacological manipulation.
- The above findings are based on single cell qPCR data, bulk and single-cell RNA-sequencing data, immunostaining, comparison of bulk RNA-seq data to Brainspan datasets from the developing cortex, comparison to single-cell RNA-seq data set to published single cell datasets from the human brain^{6,7} as well as functional and pharmacological characterization.
- We hope that the panels below (Figures 2 and 3) illustrate these points:

Figure 2. Characterization of cells generated by our neuronal differentiation protocol (from Nehme et al 2018). A) Immunostaining of differentiated neurons (20x and 40x images). B) Quantification of images in A. We used MAP2 expression to normalize quantification. 95% of neurons express NEUN, while the proliferative marker KI67 was detected in only 1% of cells

($n=1137$ cells). Most neurons expressed *CUX1* and *CUX2*, transcription factors that in the mouse cortex are expressed in layers II-III⁷. *BRN2*, another marker of upper cortical layers in mouse, was present in 50% of the population ($n=1743$ cells). By contrast, transcription factors that specify neurons in deep cortical layers, such as *CTIP2*, were virtually absent. Finally, 86% of cells expressed *TBR1*, a transcription factor widely expressed in glutamatergic neurons ($n=1269$ cells). C) Violin plots of key marker genes assayed by single cell qPCR (Fluidigm Biomark). D) Violin plots showing the expression of AMPA and NMDA receptor subunits in D28 neurons (detected by single cell RNAseq). E) Violin plots showing the expression of the AMPAR subunits in single cells at day 28. C) Pearson's correlation of bulk RNAseq data of neuronal cells over time with cortical structures of the developing human brain (Brainspan dataset).

Figure 3. Analysis of marker expression in Neuronal Progenitor Cells (from Wells et al 2018). A) Immunostaining of forebrain NPC protein markers. Scale bar = 50 μ m. B, Heat map depicting percentage of NPC marker-positive cells in NPCs from 47 different hPSC lines ($n = 4$ wells), based on quantification of protein marker expression. Data are represented as mean \pm S.D.

In summary, we have examined and published data from multiple cell lines using this protocol prior to the current work, and the identity of the neurons is the same across all cell lines. In the current study, we do not detect any differences with respect to the subtype of neurons generated (Figure 3). Additionally, cells from either 22q11.2 deletion donors or controls do not cluster separately in PCA at any of the examined cellular changes (stem cells, NPCs, or neurons) (Fig. 1h of the manuscript), providing further evidence to this point.

Figure 4. Neuronal identity is unchanged in deletion neurons.

3. To help the readers better understand the function of DE genes identified in Figure 2, the authors could perform GO analysis with the DE genes identified at different time points.

We agree with the reviewer that a GO analysis would be informative. We have in fact done this, however, at the FDR < 0.05 cutoff, which is the one we use in Figure 2, there are no GO terms that pass significance after correcting for multiple testing, likely because of power, as the number of significant genes at this cutoff is relatively small. For this reason, we have further performed GO analysis on the DE genes at the $p < 0.05$ cutoff. The result of these analyses is discussed on page 20 and summarized in Tables S11 and S12. Additionally, we have used SynGO to identify genes with synaptic annotations, as discussed on pages 18-19.

4. It is unclear why the authors compared their DE genes to genes in ASD database? There is only one patient with ASD diagnosis.

We thank the reviewer for the chance to elaborate on this point. The fact that the 22q11.2 deletion is associated with different diagnoses in different individuals, including intellectual disability in 45% of individuals, and autism in 18% of individuals⁸ prompted us to examine if genes implicated in early neurodevelopmental disorders were differentially regulated in cells from individuals with 22q11.2 deletion, irrespective of the diagnosis in our cohort, where, as the reviewer mentioned, one patient has been diagnosed with ASD, 4 patients have been diagnosed with anxiety and 11 patients have been diagnosed with intellectual disability (Figure 1e).

5. It is very confusing which DE gene list was used in Figure 4. There are a total 133 DE genes in neurons identified in Figure 2b but the authors reported using 239 of the 2864 transcripts with increased abundance for the analysis in Figure 4. The authors need to explain clearly how they curated the data for this analysis.

We thank the reviewer for giving us the opportunity to clarify this point. In Figure 2b, we have listed the differentially expressed genes that passed the more stringent significance threshold of $FDR < 5\%$ ($P_{adj} = 0.05$). In Figure 4, we have considered all genes that were upregulated in neurons at a nominal significance level ($p < 0.05$, 2864 transcripts in total), and found that 239 of them have synaptic annotations. While we had indicated these significance thresholds in the text in the earlier version, we have now also included them in the figure legends for increased clarity. We also note that these figure panels have now been revised, and the numbers are slightly changed (now 2200 transcripts in total, of which 193 have synaptic annotation in the current version of Figure 4), based on Reviewer 3's recommendation to remove the line with the shorter deletion (SCBB 1430) from the analysis.

6. The authors developed a new tool (PPItools) for analyzing protein-protein interaction networks. The authors should compare this tool to other tools like STRING and explain how it can improve the analysis.

We thank the reviewer with this comment. There are two types of analyses of protein-protein interactions (PPI) presented in the manuscript. First, the analysis of excessive connectivity between a set of top differentially expressed genes in our experiment and a gene set of known disease genes (Extended Data Fig. 4c). It is important to note several challenges that were addressed in our analysis:

- Rather than looking at protein-protein connectivity enrichment within a set of genes we are looking for connectivity between the two sets of genes. This requires adequate normalization for the size of gene sets and the connectivity (degree) of each gene within the reference network.
- Since we are working with brain-related diseases and cell types, it is important that random gene sets that we compare our results to are generated in such a way that preserves expression (e.g. NDD list of genes has the same expression distribution in iPSC/NPC/Neuronal cells as any of the random gene sets that it is compared to).

While common in the field, STRING or any other methods, to our knowledge, do not have the functionality for 'between-the-gene set' connectivity analysis and do not account for co-expression of the target genes in their random permutations to evaluate significance of the observed results.

However, we still tried our best to try to replicate our findings with STRING (Figure 5 below). We used 3 sets of genes for each cell type in STRING – disease genes (IBD, NDD or Parkinson) – *list 1*, top genes from our differential expression analysis (iPSC, NPC or Neurons) – *list 2*, and a merge of the two sets – *list 3*. By taking the difference in number of connections observed in *list 3* and the sum of connections in *lists 1 & 2* we approximated the number of connections between the disease gene set and a set of corresponding differentially expressed genes from our experiment. The resulting number of connections was normalized to a total number of nodes in *list 3* (we are unable to use proper normalization for node degrees as STRING does not provide such functionality).

The figure below nicely replicates the pattern presented in our Extended Data Fig. 4c; however, STRING does not have the appropriate functionality to evaluate significance of this observation.

Figure 5. STRING analysis.

Second, analysis of the most-weighted connected subgraph with our in-house R-package – uses the P-values of differential expression analysis to construct a weighted network of proteins and search for the most ‘up-weighted’ module. Working with weighted networks and this kind of analysis is absent from STRING or similar methods, therefore, it is impossible to have a meaningful comparison.

We made all the possible effort to analyze similar methods, but to our best knowledge, the method’s functionality proposed in our results presents a novelty to the field. We hope that our thorough permutational approach to simulations and usage of control gene sets for unrelated diseases should be convincing of statistical accuracy and biological relevance of the observed results. We have now clarified the advantages of this method on pages 16 and 17.

7. The authors stated that many genes in the 22q11.2 interval interact with JUN/FOS transcriptional pathways. Can the authors experimentally explore how it could contribute to disease risk? Can the authors rescue any phenotypes by manipulating this pathway in the isogenic deletion lines?

We agree with the reviewer that further exploring how the JUN/FOS transcriptional pathway might contribute to disease risk in the 22q11.2 deletion pathway is a very interesting and exciting next direction to pursue. While we are eager to begin exploring such experimental avenues, we believe that such experiments are beyond the scope of the current manuscript.

Reviewer #2 (Remarks to the Author):

The authors have prepared a very thoughtful and extensive revision. The authors did further experiments to validate RNAseq and proteomic hits. This revision is now suitable for publication and a significant addition to the literature.

We deeply thank the reviewer for these comments!

Reviewer #3 (Remarks to the Author):

The authors have evidently done a great deal to address many of the issues raised in the original set of reviews, and the manuscript overall has been improved.

However, the revised manuscript continues to include one iPSC cell line that has no 22q11.2 deletion. This 134 kb common variant within the low copy repeat A (LCR22-A) sequence, Karolinska sample SCBB-1430 from a 66 year old patient with schizophrenia. The fact that there are interesting and important results despite the erroneous inclusion of this cell line does support the power of the remaining sample, but for the authors not to recognize that this cell line does not belong in a “22q11.2 deletion” group, and indeed now to repeatedly call this a “nested 22q11.2 deletion”, is unacceptable.

There is another 22q11.2 deletion included (SCBB-1961, Stanford) that historically would not have been detected using a typical FISH probe, but would at least be called pathogenic using clinical microarray technology, and could be considered a “nested 22q11.2 deletion”, although the breakpoints provided indicate that it would not be included in most studies of the 22q11.2 deletion syndrome as these usually rely on the commonly deleted region that includes HIRA (TUPLE1 probe), and involve the recurrent LCR22A-LCR22B and LCR22A-LCR22C deletions that are nested within the LCR22A-LCR22D deletion region. The 2.5-3.0 Mb LCR22A-LCR22D deletion is the most common of the collection of rare, recurrent, pathogenic 22q11.2 deletions with replicated high risk of schizophrenia, and appears to be the 22q11.2 deletion extent for the other 18 cell lines used in the sample here, including the two used in their “pilot” study.

This remains an important point for the authors to address, given that the “nested 22q11.2 deletion” that is represented by many engineered mouse models is closest to the recurrent LCR22A-LCR22B human 22q11.2 deletion (not present in any of the samples used here). The manuscript would benefit from demonstration that the authors understand these basic issues; most interested readers will expect the breakpoints of the 22q11.2 deletions to be made explicit using standard nomenclature (LCR22A-LCR22D) in order to appreciate the significance of the findings.

In fact, there is some emerging evidence that smaller nested LCR22A-LCR22B 22q11.2 deletions may have a somewhat less severe neuropsychiatric phenotype, at least with respect to intellect (see e.g., Zhao et al., 2018 PMID: 30289625). The older references presented to attempt to support inclusion of a unique nested variant 22q11.2 deletion (and the erroneous non-deletion) cell line are superseded by this reference which reports data for a large enough sample to be able to detect phenotypic differences.

The remainder of the manuscript implies that there should be “...‘the’ 50% decrease in expression of ‘the hemizygote’ genes found in the (22q11.2) deletion region” (p 8).

We thank the reviewer for detailing the reasoning of why we should remove the SCBB 1430 line with the small deletion from the analysis. We have now taken the reviewer’s recommendation

and repeated all analyses in the manuscript in a dataset excluding data from this line and provided a rationale for this on page 7.

As a result, we have now changed the following figure panels: Fig 1i; Extended Data Fig 1 g, h; i; Extended Data Fig 2 a-d; Fig 2b,c,d,e; Extended Data Fig 3 a, b, c, d, e; Fig. 4a, d (now c), g(now f); Extended Data Fig 4 a, b, c, Extended Data Fig 5a, b, c, d, e; Extended Data Fig 6a, b; Fig 3 b; Fig 5h; Extended Data Fig 8b, c; Extended Data Fig 9b, c, f; Fig 3 a, b, d (now c); Added Extended Data Fig 2a; deleted Extended Data Fig 2c (already in legend of Fig 2a); Fig 3 c is now Extended Data Fig 6 d. We have further modified the following tables to reflect the removal of SCBB 1430 from the dataset: Extended Data Table1, Tables S1, S2, S3, S7, S8, S9, S10, S11, S12, and S13.

In only a few instances do we refer to a dataset that still includes SCBB1430, should this be of use to any readers interested in the effect of small deletions within the 22q11.2 locus.

On the reviewer's recommendation, we have also added the standard breakpoint nomenclature to Extended Data Table 1 and added a discussion of the Zao et al 2018 reference suggested by the reviewer, which indeed finds that the size of the deletion has a modest effect on IQ, with individuals with the shorter 1.5Mb deletion having "modestly higher IQ scores" compared to individuals with the 3Mb deletion (pages 7 and 33).

Given that 12 of the cell lines used were from individuals with schizophrenia, with three to four others too young to know outcome, this means that 12 to 16 of the 19 cell lines could be from individuals with 22q11.2 deletion and schizophrenia or related psychotic illness. Possible implications of this ascertainment bias on the findings should be presented in the Discussion.

We agree with the reviewer that the majority of patients in our cohort have been diagnosed with schizophrenia, and the possibility of ascertainment bias because of this merits to be noted in the discussion. We have now added this on page 32. We further note that our goal is to model the genetic liability to schizophrenia, and we believe that even when individual donors do not have a clinical presentation of schizophrenia, we might assume that they are at increased risk for schizophrenia because of the deletion genotype.

It would be useful to point out that Parkinson's disease, while likely to be "unrelated" to findings in excitatory neurons, is a known associated feature of 22q11.2 deletions, thus would not be considered "unrelated" to the condition under study.

We thank the reviewer for this comment. This analysis was specifically added in response to the following request from reviewer 2: "*Comparison of the PPI enrichment to an unrelated brain disorder gene list such as Parkinson's would serve as a strong additional control*". We agree that unrelated is not an appropriate word in this context, in fact we had used "unrelated" to refer to IDB. We have now clarified this and added "non-neurodevelopmental" to characterize Parkinson's (page 12).

Data and methodology

While the sections on the MAGMA and SCHEMA variant findings have been changed, I find that the authors' explanation of the findings and differences between these in the response to reviewers letter is clearer than that in the manuscript; the same for the SynGO findings.

We have now simplified the language in these sections and removed redundant analyses from the schizophrenia heritability and SynGO sections (pages 12-15 and 18-20). We note that the corresponding figures have now been changed to reflect the new dataset and analysis (excluding line SCBB 1430 with the small deletion), and that, in response to both reviewers 1 and 3's comments we have now streamlined, simplified and summarized the discussion in this section to clarify the message.

I would expect that GO (gene ontology) and PPI analyses would have very similar origins, thus it would not be surprising for results to converge.

We agree with the reviewer that it's not surprising for results from GO and PPI analyses to converge, as they indeed have similar origins, and describe biological features of the data. We, however, do note that the two approaches are not identical, and have slightly different assumptions about biological relatedness as there are also differences between the two methods: GO annotations highlight genes that are in the same pathway but don't have to physically interact. Our overall rationale was that if multiple related analyses, which are sensitive to different aspects of biological function, converge on the same findings, this would add to the robustness of the findings.

The Discussion, while improved and now containing mention of some limitations, continues to need improvements with respect to placing the study results in the context of the existing literature. For example, it is consistent with evidence from other studies that the 22q11.2 deletion confers risk for various phenotypes at least in part through pathways known from other individuals without this deletion. It is not surprising that genes (? "components") in the 22q11.2 deletion region play a role in transcriptional signals - the mouse literature is replete with evidence of this. The results of the study suggest that the 22q11.2 deletion may have an impact on pathways linked to proliferation, etc.

We thank the reviewer for this comment. We have further expanded the discussion about limitations (pages 32-33) and have now added a note about how our findings are consistent with existing literature around transcriptional regulation, on page 27.

A note about TBX1 not appearing in any of the PPI results presented would be welcome, particularly given the speculation about its role with respect to MEF2C.

We have now added this on page 27.

Clarification that the authors found some enrichment of expression of genes in the 22q11.2 deletion excitatory neurons studied that in other studies of schizophrenia without a 22q11.2 deletion show evidence of rare loss of function variants, would be welcome.

We have now further clarified this on page 29.

There is not a perfect correlation between mRNA and protein levels, as the study results support, with the likelihood that complex regulatory mechanisms are involved, including those that may be affected by decreased DGCR8 gene dosage, and not always in readily appreciated / simple directions. Would the authors consider this, and also that the strongest predictor of protein level for any given gene is the rate of translation, followed by mRNA levels (see PMIDs: 21593866, 30694346). It would seem premature to exclude any of the genes in the 22q11.2 deletion region from a role in the findings of this study (p 30, 32).

We thank the reviewer for this suggestion and have now added a discussion of this on pages 30-31.

The authors need to specify which of “the same repetitive elements” (p 31) were involved in the engineered human 22q11.2 deletion iPSCs, and relate to “this large structural variant”, and “which associations...the deletion (was) sufficient to cause” (presumably the low copy repeat sequences A and D flanking the typical 22q11.2 deletion region). This is also very important as it relates directly to the other iPSCs studied that were derived from the 18 individuals included in the study who had LCR22A to LCR22D 22q11.2 deletions. While the CRISPR/Cas9 generated lines very much implicate the importance of the effects of this 2.5 Mb 22q11.2 deletion on the rest of the genome including factors involved in NDD/ASD and schizophrenia risk, these effects would presumably remain affected by the background genome (p 31) of this male line with no phenotypic information.

We thank the reviewer for this comment. We have now specified in the discussion as well that “the same repetitive elements” that we have targeted are sequences in LCR22A and LCR22D (on page 31), in addition to the results section (page 22) and listing the gRNA sequence in LCRA and LCRD on page 62 (methods sections).

2.5 Mb of 51 Mb would not generally be considered consistent with saying “The 22q11.2 deletion lacks a large portion of chromosome 22...” even if only the most common LCR22A to LCR22D 22q11.2 deletion was being considered here.

We have now changed this statement to read: “The 22q11.2 deletion typically removes 2.5-3Mb of chromosome 22...” on page 32.

Specific discussion of the potential effects of the 22q11.2 deletion size (including as noted above), diagnosis of most originating individuals with schizophrenia, and male excess of cases, needs to be included in the limitations section. If there were data presented in the results related to age or sex, as hinted at in the Discussion (p33), I did not see these.

We have now elaborated on this on pages 32-33.

Clarity and context:

Wording in the abstract sentence beginning “In more differentiated excitatory neurons,” about the results related to indirect association needs to be clearer lest readers interpret that the variants assessed were those in the iPSCs. Throughout the manuscript, care in use of the terms “mutant” or “mutations” is warranted when the authors really mean “variants”. The same goes for “the deletion” when they mean “effects of the 22q11.2 deletion”. And (p 19) “In comparison, genes...” with reduced expression by the presence of the 22q11.2 deletion in excitatory neurons... would appear more accurate than what is stated. More judicious use of definitive statements e.g., “are” when the authors mean “may be” (e.g., p 17), and of terms such as “strikingly” would be welcome (e.g., p 14). This is not a “patient-driven study”.

We have now changed the wording in these different instances to help with clarity, as suggested by the reviewer on pages 2, 3, 5, 11, 14, 18, 19, 29.

There continues to be some evidence of lack of proof-reading and attention to detail, e.g., “in road” (p 3), “seven genes” then list only 6 (p 8), “falls shorts” (p32). There also needs to be some attention to ensure that the text and figures match with respect to gene names used (e.g., TRKA receptor, p 17). Also, in figures there is still a need to change to white font if the font is on a dark background for readability.

We thank the reviewer for suggesting these improvements. We have now corrected this in all the listed instances (on pages 3, 8, 32, 17), and proof-read the manuscript again. We have further switched the font in Figure 4a for better readability.

References:

- 1 Nehme, R. *et al.* Combining NGN2 Programming with Developmental Patterning Generates Human Excitatory Neurons with NMDAR-Mediated Synaptic Transmission. *Cell Rep* **23**, 2509-2523, doi:10.1016/j.celrep.2018.04.066 (2018).
- 2 Wells M, S. M., Piccioni F, Hill E, Mitchell J, Worringer K, Raymond J, Kommineni S, Chan K, Ho D, Peterson B, Siekmann M, Pietilainen O, Nehme R, Kaykas A, Eggan K. . Genome-wide screens in accelerated human stem cell-derived neural progenitor cells identify Zika virus host factors and drivers of proliferation *BioRxiv* (2018).
- 3 Fan, L. Z. *et al.* All-optical synaptic electrophysiology probes mechanism of ketamine-induced disinhibition. *Nat Methods* **15**, 823-831, doi:10.1038/s41592-018-0142-8 (2018).
- 4 Mitchell JM, N. J., Ghosh S, Handsaker RE, Mello CJ, Meyer D, Raghunathan K, de Rivera M, Tegtmeyer M, Hawes D, Neumann A, Nehme R, Eggan K, McCarroll SA. . Mapping genetic effects on cellular phenotypes with “cell villages”. . *BioRxiv and Cell, in revision* (2020).
- 5 Wells MF, N. J., Ghosh S, Mitchell J, Mello CJ, Meyer M, Raghunathan K, Tegtmeyer M, Hawes D, Neumann A, Worringer KA, Raymond JJ, Kommineni S, Chan K, Ho D, Peterson BK, Piccioni F, Nehme R, Eggan K, McCarroll SA. . Natural variation in gene

expression and Zika virus susceptibility revealed by villages of neural progenitor cells. *BioRxiv* (2021).

- 6 Darmanis, S. *et al.* A survey of human brain transcriptome diversity at the single cell level. *Proc Natl Acad Sci U S A* **112**, 7285-7290, doi:10.1073/pnas.1507125112 (2015).
- 7 Pollen, A. A. *et al.* Low-coverage single-cell mRNA sequencing reveals cellular heterogeneity and activated signaling pathways in developing cerebral cortex. *Nat Biotechnol* **32**, 1053-1058, doi:10.1038/nbt.2967 (2014).
- 8 Swillen, A. & McDonald-McGinn, D. Developmental trajectories in 22q11.2 deletion. *Am J Med Genet C Semin Med Genet* **169**, 172-181, doi:10.1002/ajmg.c.31435 (2015).

REVIEWER COMMENTS

Reviewer #1 (Remarks to the Author):

In the revised manuscript, the authors performed additional experiments to characterize the differentiation process and demonstrated the advantages of the newly developed PPItools for analyzing protein-protein interaction networks. The manuscript is now suitable for publication in Nature Communications.

Reviewer #3 (Remarks to the Author):

This is a study of iPSCs from heterogeneous sources, 18 with a typical pathogenic 22q11.2 deletion (6 sources; 13/18 with a schizophrenia and/or moderate intellectual disability phenotype), and 29 others deemed controls (four sources), taken to three developmental stages - day 1 (stem cells), day 4 (neural progenitor cells), and day 28 (excitatory neurons) at 4 labs, with RNA sequencing data; protein and functional experiments were confined to the day 28 cells. There are also 2 male isogenic lines with CRISPR/Cas9 engineered 22q11.2 deletions similarly studied, one of which has a full A-D deletion and the other with a nested 22q11.2 deletion that does not include the most proximal 10-12 genes in the 22q11.2 deletion region.

The authors have done some further work to address some of the issues raised in the previous two sets of reviews, but important issues remain to be improved, as highlighted below.

1. The manuscript clarity would be vastly improved by a figure showing the typical 22q11.2 deletion region with the low copy repeat regions, and the breakpoints of both the typical 22q11.2 deletions (18 from patients and 1 of the isogenic lines) and the atypical breakpoints of the SCBB-1961 (Stanford) line, and those of the isogenic line with breakpoints 18,892,575-21,460,220. The Karolinska sample SCBB-1430 with the 134 kb common variant within the low copy repeat A (LCR22-A) sequence should clearly not be used at all, nor indicated to have a pathogenic 22q11.2 deletion. Though at least some of the manuscript results now exclude this line, there remains some lack of clarity in the manuscript as a whole about this key issue.
2. The limitations in the Discussion should include the possible effects on the results (particularly for the background genetics) of including many cell lines from a sub-isolate of Northern Finland, known to have a high prevalence of intellectual disability and schizophrenia, even compared to other areas of Finland (PMID: 30679432). Also, it would be helpful to have some discussion of the fact that for other pathogenic copy number variations, there appears to have been an easier time of the engineering of isogenic lines all with the same breakpoints (e.g., 16p11.2 deletion, 15q13.3 deletion) at the Broad.
3. The Discussion as a whole continues to need attention to overall clarity of the text.
4. References throughout the manuscript could do with a review and updating, particularly with respect to the work on 22q11.2DS.
5. There are multiple errors in the Figures and Tables cited in the text.

NCOMMS-21-37746A Response to reviewers

REVIEWERS' COMMENTS

Reviewer #1 (Remarks to the Author):

In the revised manuscript, the authors performed additional experiments to characterize the differentiation process and demonstrated the advantages of the newly developed PPI tools for analyzing protein-protein interaction networks. The manuscript is now suitable for publication in Nature Communications.

We thank the reviewer for taking the time to evaluate our manuscript one last time and are delighted by their decision!

Reviewer #3 (Remarks to the Author):

This is a study of iPSCs from heterogeneous sources, 18 with a typical pathogenic 22q11.2 deletion (6 sources; 13/18 with a schizophrenia and/or moderate intellectual disability phenotype), and 29 others deemed controls (four sources), taken to three developmental stages - day 1 (stem cells), day 4 (neural progenitor cells), and day 28 (excitatory neurons) at 4 labs, with RNA sequencing data; protein and functional experiments were confined to the day 28 cells. There are also 2 male isogenic lines with CRISPR/Cas9 engineered 22q11.2 deletions similarly studied, one of which has a full A-D deletion and the other with a nested 22q11.2 deletion that does not include the most proximal 10-12 genes in the 22q11.2 deletion region.

The authors have done some further work to address some of the issues raised in the previous two sets of reviews, but important issues remain to be improved, as highlighted below.

We thank the reviewer for their thoughtful comments, which we have addressed below point by point.

1. The manuscript clarity would be vastly improved by a figure showing the typical 22q11.2 deletion region with the low copy repeat regions, and the breakpoints of both the typical 22q11.2 deletions (18 from patients and 1 of the isogenic lines) and the atypical breakpoints of the SCBB-1961 (Stanford) line, and those of the isogenic line with breakpoints 18,892,575-21,460,220. The Karolinska sample SCBB-1430 with the 134 kb common variant within the low copy repeat A (LCR22-A) sequence should clearly not be used at all, nor indicated to have a pathogenic 22q11.2 deletion. Though at least some of the manuscript results now exclude this line, there remains some lack of clarity in the manuscript as a whole about this key issue.

We have followed the reviewer's recommendation and added a new Figure panel (Supplementary Fig. 1a, Top) illustrating the deletion breakpoints in relation to the LCRs in each cell line (in addition to listing this information in Table 1).

2. The limitations in the Discussion should include the possible effects on the results (particularly for the background genetics) of including many cell lines from a sub-isolate of Northern Finland, known to have a high prevalence of intellectual disability and schizophrenia, even compared to other areas of Finland (PMID: 30679432). Also, it would be helpful to have some discussion of the fact that for other pathogenic copy number variations, there appears to have been an easier time of the engineering of isogenic lines all with the same breakpoints (e.g., 16p11.2 deletion, 15q13.3 deletion) at the Broad.

We have now added both points to the discussion (pages 28 and 29).

3. The Discussion as a whole continues to need attention to overall clarity of the text.

We have edited the discussion for brevity and clarity (pages 24-29).

4. References throughout the manuscript could do with a review and updating, particularly with respect to the work on 22q11.2DS.

We have updated the references as indeed some of the preprints that we had initially cited are now published in peer reviewed journals.

5. There are multiple errors in the Figures and Tables cited in the text.

We have confirmed that all citations of the Figures and Tables in the text are accurate and reflect the latest updates.